



# Simulation-Based Inference for Parameter Estimation of Complex Watershed Simulators

Robert Hull[1], Elena Leonarduzzi[2], Luis De La Fuente[1], Hoang Viet Tran[3,4], Andrew Bennett[1], Peter Melchior[5,6], Reed M. Maxwell[2,3,7], Laura E. Condon[1]

[1] Hydrology and Atmospheric Sciences, University of Arizona, Tucson, AZ, USA
[2] High Meadows Environmental Institute, Princeton University, Princeton, NJ, USA
[3] Civil & Environmental Engineering, Princeton University, Princeton, NJ, USA
[4] Atmospheric Sciences & Global Change Division, Pacific Northwest National Laboratory, Richland, WA, USA
[5] Center for Statistics and Machine Learning, Princeton University, Princeton, NJ, USA
[6] Department of Astrophysical Sciences, Princeton University, Princeton, NJ, USA
[7] Integrated GroundWater Modeling Center, Princeton University, Princeton, NJ, USA
*Correspondence to*: Robert Hull (roberthull@email.arizona.edu)

**Abstract.** High-resolution, spatially distributed process-based (PB) simulators are widely employed in the study of complex watershed processes and their responses to a changing climate. However, calibrating these simulators to observed data remains a significant challenge due to several persistent issues including: (1) intractability stemming from the computational demands and complex responses of simulators, which renders infeasible calculation of the conditional probability of parameters and data, and (2) uncertainty stemming from the choice of simplified model representations of complex natural hydrologic processes. Here we demonstrate how Simulation-Based Inference (SBI) can help address both these challenges for parameter estimation. SBI uses a learned mapping between parameter space and observed data to estimate parameters for generation of calibrated model simulations. To demonstrate the potential of SBI in hydrologic modelling, we conduct a set of synthetic experiments to infer two common physical parameters, Manning's coefficient and hydraulic conductivity, using a representation of a snowmelt-dominated catchment in Colorado, USA. We introduce novel deep learning (DL) components to the SBI approach, including an 'emulator' as a surrogate for the process-based simulator to rapidly explore parameter responses. We also employ a density-based neural network to represent the joint probability of parameters and data without strong assumptions about its functional form. While addressing intractability, we also show that where uncertainty about model structure is significant, SBI can yield unreliable parameter estimates. Approaches to adopting the SBI framework to cases where model structure(s) may not be adequate are introduced using a performance-weighting approach.

## 1 Introduction

Robust hydrologic tools are necessary to understand and predict watershed behaviors in a changing climate (Condon, 2022). This need is underscored by long-term drought in the American West (Williams et al., 2022), which has led to the withering of water supplies from the Colorado River (Santos and Patno, 2022), increased groundwater pumping (Castle et al.,



2014), and uncertainty about what is next (Tenney, 2022). Hydrologic simulators that represent physical processes and connections within the hydrologic cycle (Paniconi and Putti, 2015) are very commonly used tools to address these needs. These 'process-based' (PB) simulators explicitly represent hydrologic states and fluxes at multiple scales based upon physics first-principles (Fatichi et al., 2016). Watershed scientists often use PB simulators to answer 'what if' questions about behavior

of watershed snowpack, soil moisture, and streamflow in a changed future because they encode fundamental processes, and not just historical data (Maxwell et al., 2021).

The behaviors and skills of these PB watershed simulators strongly depend on spatially varying parameters (Tsai et al., 2021). Parameters represent the structure and physical properties of the hydrologic system, such as the roughness of the land surface (i.e., Manning's Coefficient, M) or the water-transmitting properties of the subsurface (i.e., Hydraulic

Conductivity, K). There are many approaches to parameter determination in hydrology (Beven and Binley, 1992.; Gupta et al., 1998; Bastidas et al., 1999; Hunt et al., 2007; Vrugt and Sadegh, 2013; White et al., 2020; Tsai et al., 2021). The variety of approaches and long history of research in this area underscores that there is "no obvious formulation of [parameter determination] that previous generations of modelers have overlooked" (Hunt et al., 2007). Yet, the question of how best to infer parameters for PB simulators remains unsettled.

Parameter determination remains a challenge with watershed PB simulators, and an impediment to robust, physics-informed hydrologic predictions. There are two related and ongoing difficulties that make parameter determination a very challenging problem. The first is the problem of intractability. For a dynamical watershed simulator with a range of possible model configurations, many combinations of parameters may be plausible given observed data (Beven, 2011; Nearing et al, 2015). Therefore, many have argued it may be preferable to simulate distributions of hydrologic variables and the underlying

parameters that give rise to them (e.g. Vrugt and Sadegh, 2013). Intractability arises when these distributions cannot be approximated for theoretical or computational reasons. For example, large-scale, high-resolution PB simulations can require massively parallel, high-performance computing (e.g., Maxwell et al., 2015), limiting the number of exploratory model runs due to computational demands. A solution to the problem of intractability needs to efficiently approximate complex distributions of probable parameters given observations with a sufficient level of accuracy and precision.

Deep learning (DL) may provide new opportunities vis-à-vis the intractability problem in parameter determination. In DL, behaviors are learned from data, as opposed to PB approaches, which derive behavior from established theory. The Earth Sciences have recently seen greater adoption of DL approaches (Wilkinson et al., 2016), for example in streamflow prediction (Kratzert et al., 2018). However, DL methods are not widely used in watershed prediction due to the "inadequacy of available data in representing the complex spaces of hypotheses" (Karpatne et al., 2017), such as watershed observations.

Recently, there has been a push for methods that can incorporate process understanding into DL models (e.g., Zhao et al.,





2019; Jiang et al., 2020). Still, studies are rare that employ DL to improve PB simulator performance by aiding in the hunt for better parameters[1] (e.g., Tsai et al., 2021).

Simulation-based inference (SBI) is a DL-informed approach to PB parameter determination that has shown promise in particle physics (Cranmer et al., 2020), cosmology (Alsing et al., 2019), and neural dynamics (Lueckmann et al., 2017). In SBI, a neural network is employed to approximate the conditional density of parameters and simulated outputs from the behavior of a simulator. That conditional relationship can be evaluated using observations to estimate a set of probable parameters. Surrogate simulators are neural networks that emulate the complex interdependence of variables, inputs, and parameters encoded in PB simulators, such as watershed simulators (Maxwell et al., 2021; Tran et al., 2021). Once trained, surrogate simulators can preserve fidelity to the PB simulator, run at a fraction of the cost, and speed up the exploration of parameter space. Restated, this approach uses one neural network (the 'surrogate') to quickly generate thousands of simulations that are utilized to train another neural network (via conditional density estimation) to develop a statistical representation of the relationship between parameters and simulated data. Via SBI, this statistical representation can be used to infer distributions of PB parameter values based on observed data. Assuming the model is correctly specified, the inferred set of parameters accurately and precisely reflects the uncertainty of the parameter estimate (Cranmer et al., 2020). To our knowledge, applications of SBI in hydrology have been limited (e.g., Maxwell et al., 2021). A brief introduction to SBI is presented in the background section.

A second challenge to parameter determination is the problem of epistemic uncertainty arising from limited knowledge, data, and understanding of complex hydrologic processes. The sources of epistemic uncertainty in the modelling process are various, including: uncertainties in data (for example, in model inputs and misleading information in observed data used to train and assess models); uncertainties derived from performance measures and information to omit; and uncertainties about model structure, which arise from the inherent challenge of choosing simplified models to represent complex processes (Leamer, 1978; Beven & Binley, 1992; Draper,1995; Gupta et al., 2012; Nearing et al., 2015). In this work, we focus on a subclass of epistemic uncertainty of appropriate model structure(s) known as "model misspecification", in which a unique and optimal model structure is assumed to exist but is unknown. Discounting the role of uncertainty about appropriate model structures can have profound consequences on the insights we draw from inference tasks like parameter determination .

A common challenge is the potential under-representation of uncertainty stemming from the choice of model structure. This issue becomes evident when inference approaches yield parameter estimates that are overly confident, which can be problematic when a more conservative estimate that accounts for the inherent uncertainties about model structure is preferred (Beven, 2011; Cranmer, 2020; Hermans, 2021). One potential remedy is the use of ensemble modelling, where multiple model structures are employed to capture a range of plausible behaviours. The challenge then becomes deciding upon which model structures to consider and how to combine them. Generalized Likelihood Uncertainty Estimation, or GLUE

---

[1] We make a distinction between the parameters of PB simulators and the parameters embedded in neural networks, which are optimized during training by backpropagation. In this report, we almost-exclusively refer to the parameters of PB simulators even as we discuss the capacity of neural networks to learn and represent them.





(Beven and Binley, 1992; Beven and Binley, 2014), associates a measure of belief with each selected model structure and configuration, forming a conceptually simple way of weighting ensembles of predictions to estimate uncertainty stemming from various sources. A similar principle underlies Bayesian Model Averaging, or BMA (Leamer, 1978; Hoeting et al, 1999;
Raftery et al., 2005; Duan et al., 2007). While GLUE and BMA differ in their implementations, they both adhere to the principle that models exhibiting behaviours closely aligned with observations should hold stronger credibility and carry greater significance within an ensemble of models; and non-behavioural model structures should be assigned a low probability or rejected. In the case of GLUE, this measure of credibility is derived from a modeler's choice of metric, or informal likelihood function (e.g. Smith et al, 2008). GLUE and BMA are further described in the background section.

105       The primary objective of this work is to demonstrate an approach to generating accurate and precise estimates of the spatially distributed parameters of a hydrologic simulator where conventional methods might struggle due to the intractability problem. A secondary goal is to explore how this workflow could be extended to yield meaningful parameter estimates considering uncertainty about model structural adequacy. Surrogate-derived SBI is utilized to address the problem of intractability in complex parameter spaces using a statistical, deep-learning approach. The problem of model structural
adequacy is confronted using a quasi-BMA approach that utilizes an informal likelihood to weight the credibility of parameter estimates from SBI.

      We primarily use synthetic test cases with diagnosable degrees of uncertainty to test the performance of the inference workflow. Here, we determine the physical parameters of a headwater subbasin of the Upper Colorado River Basin by calibrating a PB watershed simulator to historical streamflow observations. We utilize SBI in tandem with a Long Short-Term
Memory (LSTM) surrogate for the PB simulator ParFlow (Jones and Woodward, 2001; Maxwell and Kollet, 2006; Maxwell et al., 2015a) to rapidly generate probable configurations of Hydraulic Conductivity (K) and Manning's Coefficient (M). Furthermore, we use the inferred distribution of parameters to generate streamflow predictions. We explore the influence of observed data on parameter inference with a set of experiments that systematically vary the degree of uncertainty associated with how synthetic and real observations relate to the simulator (i.e., misspecification). In the latter experiments, a form of
BMA is utilized to improve robustness of the parameter estimates to misspecification, in the extreme case by assigning zero probability to all models in the set. The experiments are outlined in Section 3.1.

      Novel aspects of the present analysis that bear noting include: (1) the usage of DL in conjunction with a PB watershed simulator to improve its performance; (2) the novel application of density-based SBI to the domain of hydrology; and (3) the usage of informal likelihood measures to directly assign model probabilities to parameter estimates made by SBI in a manner
similar to BMA. The significance of this work is to develop a framework to tackle harder inference problems in watershed modeling, and other domains of the Earth Sciences where complex PB simulators are used.



## 2 Background of inference-based approaches to hydrologic parameter determination

This section provides a brief background of methods used for parameter determination in watershed modeling and
related problems. We provide context relevant to understanding the "point of convergence" (Cranmer et al., 2020) we call
simulation-based inference (SBI), and how it is similar to and different from some other approaches to inference. We start with
a general overview of inference. Next, we discuss the traditional formulation of the inference of parameters using Bayes'
theorem (section 2.1). We then introduce what sets SBI apart from these traditional approaches (section 2.2). Next, we discuss
the role of machine learning in SBI (section 2.3). Finally, we introduce some approaches to parameter estimation under
epistemic uncertainty that have been applied in hydrology (2.4). We define 'inference' as using data (observations) and a
simulator to describe some unobserved characteristic of the system we are interested in (Cranmer et al., 2020; Wikle and
Berliner 2007).

### 2.1 Bayesian inference

Bayesian inference is a common method to extract information from observations. The essence of this formulation of
inference unfolds in three steps (Wikle and Berliner, 2007): (1) Formulate a 'full probability model', which emerges from the
joint probability distribution of observable and unobservable parameters; (2) Infer the conditional distribution of the parameters
given observed data; (3) Evaluate the fit of the simulator (given parameters inferred in step 2) and its ability to adequately
characterize the process(es) of interest.

Traditionally, to tackle inference problems we apply Bayes' Theorem. For illustration, let $\theta$ denote unobserved
parameters of interest (such as Hydraulic Conductivity); and let $Y$ represent simulated or observed data of the variable of
interest (such as streamflow). The joint probability $p(\theta, Y)$ can be factored into the conditional and marginal distribution by
applying Bayes' Rule, such that we obtain:

$$p(\theta|Y) = \frac{p(Y|\theta)\, p(\theta)}{p(Y)} \tag{1}$$

Where,

● The *data distribution*, $p(Y|\theta)$, is the distribution of data given unobservable parameters. This distribution is referred
to as the likelihood when viewed as a function of θ for a fixed Y. The likelihood function of "implicit" simulators
(such as those used in watershed modeling) is often regarded as 'intractable' – i.e., its form cannot be evaluated
(integrated), at least not in a computationally-feasible way (Cranmer et al., 2020).

● The *prior distribution, $p(\theta)$*, is our *a priori* understanding of unobservable parameters. The prior often results from a
choice made by the domain expert. For example, in watershed modeling the prior distribution arises from a belief
about the possible structures and magnitudes of parameters (for example, hydraulic conductivity) in a study domain,
as well as the probability that they could be observed.



- The *marginal distribution*, *p(Y)*, can be thought of as a normalizing constant or 'evidence'. In practice, this distribution is rarely computed as it contains no information about the parameters. As such, we do not include *P(Y)* and instead work with the unnormalized density provided by Equation 2:

$$p(\theta|Y) \propto p(Y|\theta)\,p(\theta) \qquad (2)$$

- The *posterior distribution*, *p(θ|Y)*, which is the distribution of unobservable parameters given the data. The posterior is the primary goal of Bayesian inference; it is proportional to the product of our prior knowledge of parameters and the information provided in our observations.

Inference conducted using a Bayesian paradigm has a long history in computational hydrology (Vrugt and Sadegh, 2013). However, applications have been somewhat limited due to challenges centering on the intractability of the data distribution, *p(Y|θ)*, for watershed simulators with many parameters.

## 2.2 Simulation-based inference

SBI is a set of methods that attempt to overcome the intractability of the data distribution by learning the form of the posterior distribution directly from the behavior of the simulator itself (Tejero-Cantero et al., 2020). The classic approach is Approximate Bayesian Computation (ABC), which compares observed and simulated data, rejecting and accepting simulation results based on some distance measure (Fenicia et al., 2018; Vrugt and Sadegh, 2013; Weiss and von Haeseler, 1998). While this approach has been widely used, it suffers from a range of issues, including poor scaling to high-dimensional problems (resulting in the need for summary statistics), and uncertainty arising from the selection of a distance threshold (Alsing et al., 2019). Additionally, in traditional ABC it is necessary to restart the inference process as new data become available (Papamakarios and Murray, 2016), making it inefficient to evaluate large numbers of observations (Cranmer et al., 2020).

SBI methods predicated on density estimation enable an alternative that does not suffer from the same shortcomings of ABC. The density estimation approach aims to train a flexible density estimator of the posterior parameter distribution from a set of simulated data-parameter pairs (Alsing et al., 2019). Some of the key advantages of a density estimation approach over ABC: (a) it represents the posterior[2] distribution parametrically (as a trained neural network) that can be reused to evaluate new data as it comes available; (b) it drops the need for a distance threshold by targeting an 'exact' approximation of the posterior; (c) it more efficiently uses simulations by adaptively focusing on the plausible parameter region (Papamakarios and Murray, 2016).

One general purpose workflow that we employ in this paper uses a neural density estimator to learn the distribution of streamflow data as a function of the physical parameters of the simulator and employs active learning algorithms to run simulations in the most relevant regions of parameter space (Alsing et al., 2019; Lueckmann et al., 2017). The SBI workflow is further described in Sect. 3.5, and the neural density estimator is described in Sect. 3.6.

---

[2] We share the literature's tendency to use 'conditional' and 'posterior' density interchangeably; denotations of $p(\theta \mid Y = Y_{True})$, for the posterior density evaluated at an observation $Y_{Obs}$; and $p(\theta \mid Y)$, for conditional density representative of a large set of simulated $\{\theta, Y\}$, are used when possible to reduce ambiguity.





## 2.3 The role of Machine Learning in SBI

Due to advances in the capacity of neural networks to learn complex relationships, we can learn high-dimensional
probability distributions from data in a way that was hardly possible before (Cranmer et al., 2020). This has led to strong claims
in other fields, including cosmology and computational neuroscience, regarding the potential of SBI to "shift the way
observational [science] is done in practice" (Alsing et al., 2019). While our implementation is described in more detail
throughout the methods section, we direct readers to the literature for a broader (Cranmer et al., 2020) and deeper
(Papamakarios and Murray, 2016) understanding of density based SBI.

Learning the full conditional density $p(\theta|Y)$ requires many simulated parameter-data pairs: thousands (or hundreds of
thousands) of forward simulations. This presents a challenge with some high-resolution PB simulators, where each forward
simulation can take hours of compute time to run. Many have noted that deep-learned surrogate simulators can help; after an
initial simulation and training phase, these simulators can be run forward very efficiently. "Surrogate-derived approaches
benefit from imposing suitable inductive bias for a given problem" (Cranmer et al., 2020). In our case, this "inductive bias" is
applied by learning the rainfall-runoff response of our PB domain using a Long Short-Term Memory (LSTM) model, a type
of neural network that is suited for learning temporal patterns in data (Kratzert et al., 2018). The surrogate simulator is
described in more detail in Sect. 3.3. Surrogate simulators can be used directly in the construction of viable posterior
distributions of physical parameters and run at low-cost relative to the PB simulator.

It should be noted that inference is always done within the context of a simulator (Cranmer, 2022). As such, if the
structure of the model underlying the simulator is not adequate, it will affect inference in undesirable ways. Model structural
inadequacy arises in the case when a simulator does not capture the behavior of the dynamical system, giving rise to mismatch
between simulated and observed data (Cranmer et al., 2020). SBI conducted with structurally inadequate models can result in
overly precise and otherwise erroneous inference. Similar concerns about the quality of inference arise from other potential
sources of epistemic uncertainty in the modeling process, such as undiagnosed error in the data used to condition the model.

## 2.4 Multi-model averaging and parameter determination in hydrology

Bayesian Model Averaging (BMA) is an approach developed in the literature on linear regression (Madigan and
Raftery, 1994) to address the problem of model selection. The principle is that basing inferences on one model structure alone
is risky (Hoeting et al, 1999), since "part of the evidence is spent to specify the model" (Leamer, 1978, page 91). In its simplest
form, BMA is a method of averaging the opinions of two or more competing model structures about a quantity of interest
(Roberts, 1965). In dynamical systems modeling, this approach has been adopted to create weighted averages of forecasts
derived from multiple types of models (i.e. Raftery et al, 2005). For example, BMA has been used to generate streamflow
forecasts from multiple types of rainfall-runoff models (Duan et al, 2006). Results from these analyses shows that the weighted
combination of models results in more accurate inference and descriptions of uncertainty than those derived from any one
model structure.





In BMA, consider $Y_{obs}$ to be observed data, such as a streamflow time series; a set of models $M_1,...,\ M_k$ with shared or different underlying structures; and a quantity of interest $\triangle$ to be inferred, such as a predicted variable or underlying set of parameters $\theta$. The probability of $\triangle$ in the presence of $Y_{obs}$ can be represented as a weighted average, such that:

$\quad p(\triangle\,|Y_{obs}) = \sum_{k=1}^{K} p(\triangle\,|M_k, Y_{obs})\, w_k$ (3)

Where:

- $p(\triangle\,|M_k, Y_{obs})$ is the posterior distribution of $\triangle$ given the model under consideration $M_k$ and $Y_{obs}$, which can be interpreted as the conditional probability of $\triangle$ given that $M_k$ is the best model in the ensemble (Raftery et al, 2005), and

- $w_k$ is the posterior model probability, or the model weight. This can be interpreted as the posterior probability that model $M_k$ is the best one (Raftery et al, 2005)

Even in relatively simple test cases (i.e. Raftery et al., 1997), the calculation of $p(\triangle\,|Y_{obs})$ is difficult due to the large number of possible models and computational and conceptual challenges related to $w_k$, and so defensible approximation methods are required (Hoeting, 1999). In the arena of dynamical systems modeling (i.e. Raftery et al, 2005; Duan et al, 2006),

this problem has typically been solved iteratively as an expectation-maximization problem that simultaneously maximizes the likelihood of both $p(\triangle\,|M_k, Y_{obs})$ and $w_k$, though other approaches have been employed in other domains (i.e. Ker and Liu, 2020).

Generalized Likelihood Uncertainty Estimation (GLUE) is an approach to uncertainty estimation with wide use in hydrology (Beven and Binley, 2014). GLUE recognizes that discrepancies between observed and model-simulated data often

exhibit non-random patterns, reflecting the presence of heteroscedasticity and autocorrelation resulting from errors in model structure, inputs, and data (Beven, 2012). To account for these uncertainties, GLUE allows the modeler to assign a "measure of belief" to each simulation result, reflecting their confidence in its validity. This measure of belief, or likelihood function, may not be formal in the statistical sense but serves to express the modeler's subjective judgement (Beven, 2012). The selection of an appropriate likelihood is crucial, often relying on performance metrics such as Nash Efficiency (NSE), but its choice

depends on the study objective (Smith et al., 2008). Likelihoods are used to develop acceptability limits and weight a set of acceptable models and approximate the uncertainty associated with the inference of parameters or other model-derived quantities. By considering multiple plausible model structures and developing a clear metric by which models are evaluated, GLUE provides a holistic and flexible framework for parameter estimation in the presence of uncertainty about the appropriate model structure and other epistemic uncertainties (Beven, 2012).

The current analysis adopts a strategy that combines SBI with informal likelihood weighting to address model misspecification. This approach involves generating weighted averages of estimated parameter distributions from multi-model ensembles using a form of Bayesian Model Averaging (BMA) (Eqn. 3). Specifically, we take the weighted average of the conditional estimates for $p(\theta|Y)$ (Eqn. 2) obtained through SBI for a set of rainfall-runoff simulators $M_1,...,\ M_k$, so that a range





of models and parameter combinations are considered. As in GLUE, the weights are derived from a selected performance
metric, reflecting the suitability of predictions given observed data; where performance is below a pre-defined limit of
acceptability, the model is not considered in the weighting process. The claim is that this method of model combination
mitigates over-confident inference due to model structural inadequacy without diluting the valuable information in the
parameter estimates made by SBI. The broader implication is an approach to extend the usage of SBI to situations where we
are uncertain about the appropriate model structure. We believe that being able to extend SBI in this way could, broadly
speaking, be part of a strategy to build more comprehensive understanding of the inherent uncertainties associated with
hydrological modeling approaches. Experiment 4 evaluates whether BMA produces more accurate parameter estimates and
realistic parameter spreads compared to standalone SBI. For detailed implementation specifics, refer to Section 3.8.

## 3 Materials and Methods

This section describes our implementation of surrogate-derived SBI, and four experiments undertaken to test it. We
first introduce those experiments, and the goals associated with them (Sect. 3.1). Then, we describe the domain of interest, the
Taylor River watershed (Sect. 3.2). The rest of the methods subsections describe the components, implementation, and
validation of SBI, as outlined in Table 1.

**Table 1. Outline of the components described in the methods section.**

| Section | Name | Description |
|---------|------|-------------|
| 3.1 | Experiments | |
| 3.2 | Taylor River Basin | Domain of study |
| 3.3 | ParFlow | Process-Based simulator |
| 3.4 | Long-Short Term Memory (LSTM) Network | Surrogate simulator |
| 3.5 | Simulation-Based Inference (SBI) | Method for parameter inference |
| 3.6 | Conditional Density Estimator, $q_\phi(\theta\|Y)$ | Learns distribution of parameters |
| 3.7 | Posterior Predictive Check | From inferred parameters, make prediction |
| 3.8 | Calculation of Model Weights | Method for model combination |



| 3.9 | Evaluation Metrics | Assess performance of SBI |
|-----|--------------------|--------------------------|


Figure 1 shows how the components of surrogate-derived SBI interrelate. In Fig. 1A, a small set of process-based simulations are generated by ParFlow. A LSTM neural network learns from these simulations to mimic the behavior of ParFlow, interpolating the relationship between climate forcings, watershed parameters M and K and output streamflow time series. The LSTM can be used as a ParFlow surrogate to quickly explore the streamflow response to different parameter configurations and forcing scenarios.

We leverage the efficiency of the surrogate to conduct SBI on parameters, as depicted by Fig. 1B. Our goal with SBI is to estimate probable values for the watershed parameters M and K given the occurrence of a particular streamflow observation. To that end, we randomly sample many (n=5000) parameter configurations from a prior distribution $p(\theta)$ and from the LSTM simulate an equivalent number of streamflow timeseries $Y$. This set of simulated parameter-data pairs is used to train a neural density estimator $q_\phi(\theta|Y)$, which is a deep-learned model of the full conditional density of parameters given data $p(\theta|Y)$. Once trained, the neural density estimator is evaluated with a given observation to produce a distribution of parameters, the posterior distribution $p(\theta \mid Y = Y_{Obs})$, which represent our 'best guess' of what the parameters should be.

Finally, a predictive check (Fig. 1C) ensures that the parameter estimates generate a calibrated model. The simplest version of this check is to put the estimates of parameters from the previous step back into the LSTM, which generates a new ensemble of streamflow simulations. The simulations should resemble the observation closely if the simulator captures the behavior of the dynamical system well, and parameter inference was done correctly. Optionally, the parameter estimates may be weighted using a performance evaluation of the predictive check.

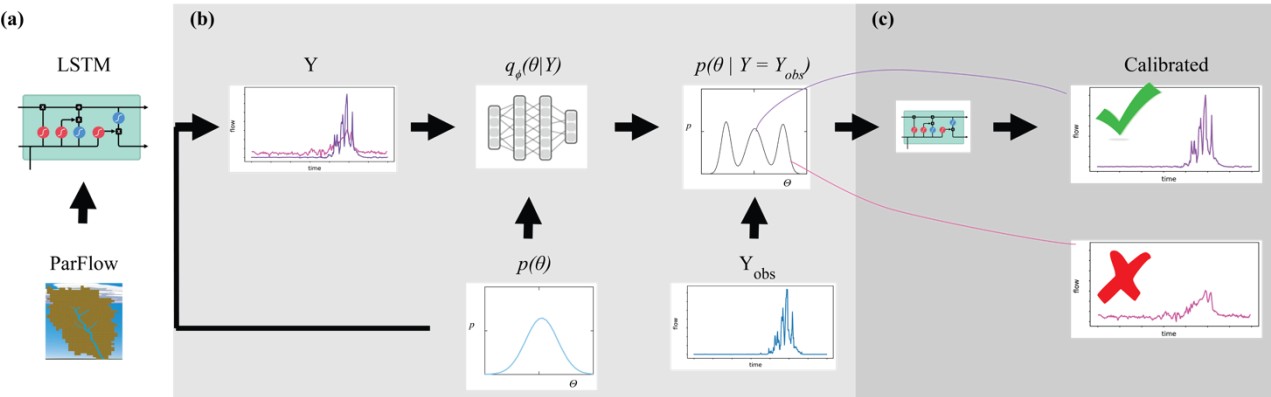


**Figure 1. An illustration of surrogate-derived simulation-based inference (SBI). In subplot (a), a Long Short-Term Memory (LSTM) neural network learns watershed behavior from ParFlow, a process-based simulator. The implementation of SBI is shown in subplot (b), where the objective is to estimate watershed parameters $\theta$ given an observation $Y_{obs}$. This parameter estimate is formally known as the posterior parameter distribution $p(\theta \mid Y = Y_{Obs})$. We randomly sample many parameter configurations from a prior distribution $p(\theta)$ and from the LSTM simulate an equivalent number of streamflow timeseries $Y$. This set of simulated parameter-**





data pairs is used to train a neural density estimator $q_\phi(\theta|Y)$. Subplot (c) shows the posterior predictive check, which involves using the parameter estimate to (ideally) generate a calibrated model.

### 3.1 Experiments

We explore the performance of SBI in four experiments. These experiments test the ability of SBI to accurately and

precisely estimate parameters for simulator calibration. The subject of interest in these experiments is the potential mismatch between observations and the simulator. To test this, we vary the degree of uncertainty associated with how observations relate to the simulator and strategies to address this mismatch. Synthetic observations with known parameters are used to conduct the experiments because they are easier to benchmark. These experiments are further described below and in Table 2, and the results are explored in Sect. 4:

1. 'Best Case': Find $p(\theta \mid Y = Y_{Obs\_LSTM})$. We use as observation the streamflow generated by a surrogate simulator (e.g., with a given combination of parameters) and use SBI to infer the parameters. Because we are treating the simulator as observations in this case (i.e. we assume the simulator can by definition generate data identical to the observation), no uncertainty exists about the structural adequacy of the model represented by the surrogate simulator. This experiment serves as a baseline check for our SBI workflow.

2. 'Tough Case': Find $p(\theta \mid Y = Y_{Obs\_ParFlow})$. We use a ParFlow simulation as observation and use SBI to infer the values of the parameters. As there is a slight mismatch between observed (in this case ParFlow simulation) and simulated data (i.e. the surrogate simulator), there is some uncertainty about the structural adequacy of the model represented by the surrogate simulator. This experiment tests whether the proposed framework, where SBI is carried out with the surrogate simulator, can be successful given misspecification of the surrogate.

3. 'Boosted Case': Find more accurate $p(\theta \mid Y = Y_{Obs\_ParFlow})$. Building from the 'Tough Case', we again use a ParFlow simulation as observation but instead use an ensemble ('boosted') surrogate simulator to infer the known parameters. Unlike in the 'Tough Case', multiple forms of the surrogate simulator are considered to represent uncertainty about the appropriate model structure. In this case we're testing whether the proposed framework can be made more robust to surrogate misspecification if multiple surrogate structures are combined.

4. 'Weighted Case': Find Bayesian Model Averaged $p(\theta \mid Y = Y_{Obs\_ParFlow}, w)$. Building from the 'Boosted Case', we add a performance measure (e.g. informal likelihood) to emphasize ('weight') credible and reject implausible forms of the surrogate simulator that have been identified by SBI. Unlike in the 'Boosted Case', uncertainty about the adequacy of surrogate simulator structures and configurations is explicitly evaluated using the likelihood weighting. This experiment tests whether the proposed framework is more robust to surrogate misspecification if competing

surrogate structures are weighted based on the fit between simulated and observed data.

**Table 2. The four experiments explore how the observation and simulator type affects the quality of parameter inference.**

| Experiment # | Name | Goal |
| --- | --- | --- |





| 1 | *Best Case* | Infer parameters given no mismatch between observed and simulated data |
| 2 | *Tough Case* | Infer parameters given some mismatch between observed and simulated data |
| 3 | *Boosted Case* | Infer parameters given some mismatch between observed and multi-model simulated data. |
| 4 | *Weighted Case* | Infer parameters given some mismatch between observed and simulated data from multiple models weighted on by their goodness of fit. |

## 3.2 Taylor River – The Domain

The physical area of study is the Taylor River headwater catchment located in the Upper Colorado River Basin (Figure 2). The Taylor is an important mountain headwater system for flood control and water supply in the Upper Colorado River Basin (Leonarduzzi et al., 2022). This catchment is at an elevation of between 2451 and 3958 meters above mean sea level and has a surface area of around. 1144 $km^2$. This catchment is snowmelt-dominated in summer. The geographical extent of the watershed is defined by the USGS streamflow gage in Almont, Colorado (ID: 09110000) at the basin outlet. Over the full

period of record (1910 - 2022), the lowest average monthly discharges are recorded in January and February, with values of approximately 100 [cfs] (3 [cms]), after which there is a steady increase of discharge and generally wetness in the catchment up until June, when an average discharge of approximately 900 [cfs] (25 [cms]) is recorded. Synthetic data corresponding to the Almont gage (USGS 09110000) location are used for Experiments 1-4, as described in Sect. 3.1. Observed streamflow data from water year 1995 are revisited in the discussion and Appendix E.






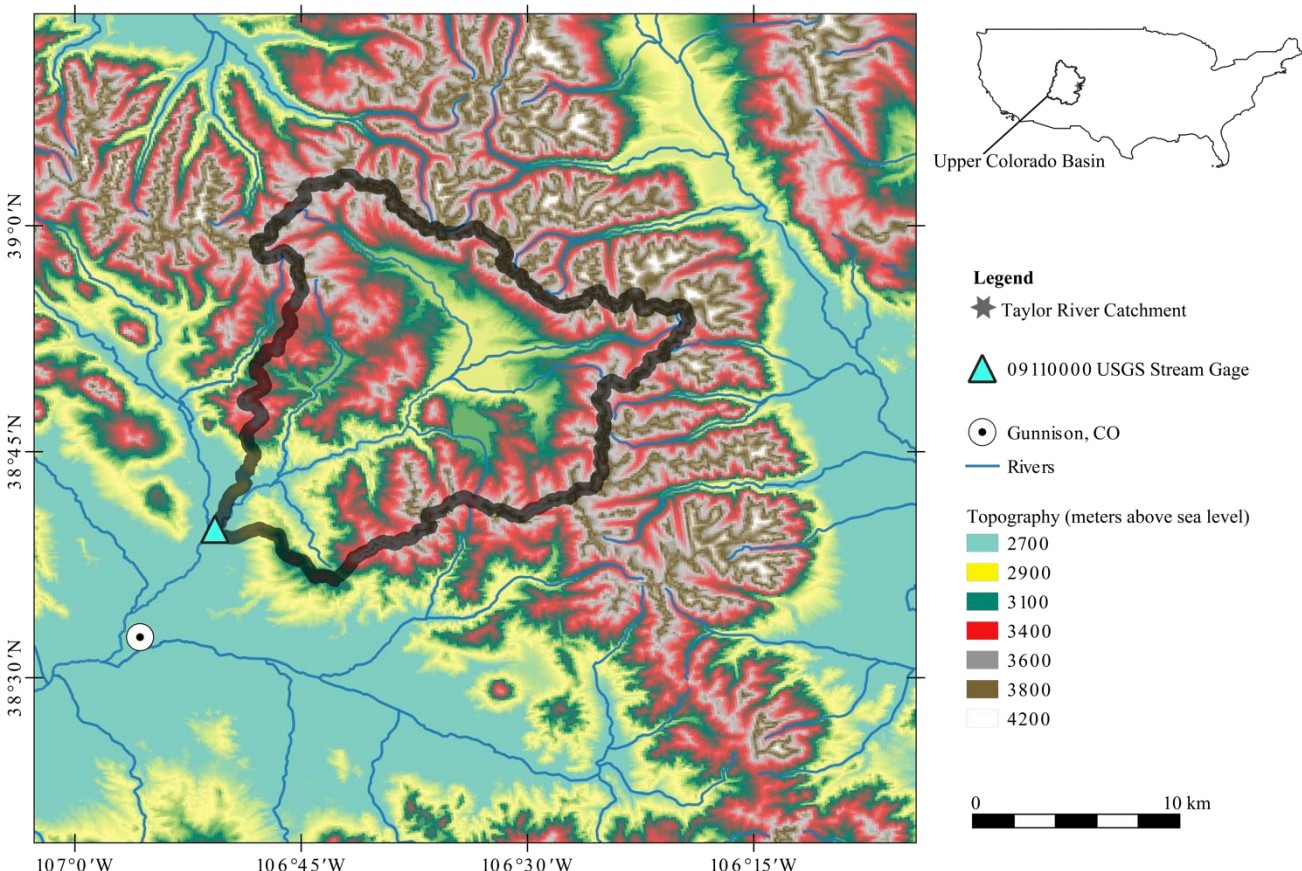

**Figure 2. Map showing the study domain Taylor River catchment near Almont, Colorado.**

### 3.3 The Process-Based Simulations (ParFlow)

We use the integrated hydrologic model ParFlow-CLM to simulate groundwater and surface water flow in our domain. ParFlow-CLM is designed to capture dynamically evolving interactions between groundwater, surface water and land surface fluxes (Jones and Woodward, 2001; Maxwell and Kollet, 2006; Maxwell et al., 2015a). In the subsurface, variably saturated flow is solved using the mixed form of Richards Equation. Overland flow is solved by the kinematic wave approximation and Manning's equation. ParFlow is coupled to the Common Land Model (CLM). CLM is a land surface model

which handles the surface water-energy balance (Maxwell and Miller, 2005; Kollet and Maxwell, 2008). It is thus well-suited to examine evolving watershed dynamics at the large scales (e.g., Maxwell et al., 2015b), as in the Taylor River Basin in Colorado, USA.



The Taylor catchment is represented by ParFlow at 1 kilometer resolution, with five vertical layers of total depth 102 meters (Leonarduzzi et al., 2022). As with Leonarduzzi et al., 2022, all the required input files - including soil properties, landcover, and meteorological forcings - are subset from Upper Colorado River Basin ParFlow-CLM simulations of Tran et al. 2022. The subsurface contains 23 separate soil and geological units.

We explore the sensitivity of streamflow to a large ensemble of different configurations of Manning's roughness coefficient (*M*), and hydraulic conductivity (*K*). For the baseline configuration of the model, *K* ranges between 6.16e-03 and 2.69e-01 [m/h] across the 23 spatial units; *M* is constant across the domain surface at 2.4e-06 [h/m^(1/3)]. An ensemble of 183 simulations is generated by systematically varying *M* and *K*. For *M* since the values are spatially constant it is easy to adjust this single value. *K* is spatially variable; therefore, we apply a single scaling factor to all three dimensions (Table A1). To make the distinction clear, we call these 'single' scalar representations $K_s$ and $M_s$, respectively. The values $K_s$ and $M_s$ used in this study are shown in Table A2. A sensitivity analysis of streamflow to parameter configurations is shown in Fig. A1.

All simulations are run for a one-year period (8760 hours) using forcings from water year 1995 taken from Tran et al., 2020. Surface pressure outputs are converted to runoff using the overland flow utility built into ParFlow. This study focuses on runoff at the cell closest to USGS gage 09110000. We convert to cubic feet per second (cfs) for direct comparison to gaged data and rescale from 0 to 1. Streamflow simulations from ParFlow are relatively more sensitive to changes in K than M, as shown in Fig. A1. The purpose of generating this ParFlow ensemble is not to create the most diverse set of system realizations but provide a foundation from which to train the surrogate model and test performance of the simulation-based inference approach.

## 3.4 The Surrogate Simulator (LSTM)

We employ a Long Short-Term Memory (LSTM) network to learn from our process-based simulator ParFlow. LSTM networks are neural networks that are designed to learn temporal relationships (Rumelhart et al., 1986; Hochreiter and Schmidhuber, 1997). They have had some use for prediction in hydrology (Kratzert et al., 2018) to learn how sequences of previous meteorological forcing data affect streamflow at the basin outflow. In our study, an LSTM network learns the response of streamflow at gaged location 09110000 to forcings and parameters in the Taylor River basin, as defined by the ensemble of ParFlow simulations described in Sect. 3.3.

Throughout our experiments, we use an LSTM with 10 input features containing forcings *X* and parameters *θ*, and one output class containing streamflow *Y*. As in Kratzert et al. 2018, we employ a 'look-back' approach. For each sample, the LSTM ingests a sequence length of '*l*'=14 days of previous forcings weighted by scalar representations of ParFlow parameters ($K_s$, $M_s$) and returns streamflow the next day. More explicitly:

$$Y_{t+1} = LSTM(X_{t \to (t-1)}, K_s, M_s) \qquad (4)$$

where $Y_{t+1}$ is the streamflow the next day, *l* is the 'look back' which controls the length of the input sequence used for prediction, $X_{t \to (t-1)}$ are vectors containing sequences of forcing data from today (i.e., day *t*) back to day *t* minus *l* for each of the 8 forcing variables. $K_s$ and $M_s$ are scalar representations of the ParFlow parameters hydraulic conductivity (*K*) and





Manning's roughness (*M*). Since these values do not vary over time each is ingested as a vector repeated *'l'* times by the LSTM.

The relevant hyperparameters used to fit the LSTM surrogate are further defined in Table A1 and B1. Fig. B1A shows the distribution of train-validation and test sets across parameter space and the performance of the LSTM relative to ParFlow

on a streamflow time series generated by a randomly selected test parameter set, $\theta_A$. $\theta_A$ is used throughout the results section for benchmarking. The LSTM captures the general streamflow behavior quite well, but not quite perfectly (Figure B1B). We emphasize that the goal here is to produce a surrogate simulator adequate for the simulation-based inference of parameters $K_s$ and $M_s$.

### 3.5 Implementation of Simulation-Based Inference

The goal of SBI is to infer appropriate values flexibly and efficiently for simulator parameters, given a particular observation. SBI is illustrated in Fig. 1B. Take $\theta$ to be a vector of parameters that control a simulator, and let *Y* be a vector of simulated data. The simulator implicitly defines a conditional probability *p(Y|θ)*, which may very well be analytically intractable. *p(θ)* encodes our prior beliefs about parameters. We are interested in inferring the parameters $\theta$ given an observation $Y_{Obs}$, i.e., we would like to know the posterior probability density $p(\theta|Y=Y_{Obs})$, after Papamakarios and Murray

400  (2016):

$$p(\theta|Y = Y_{Obs}) \propto p(Y = Y_{Obs}| \theta)\, p(\theta) \tag{5}$$

where $\theta$ contains $K_s$ and $M_s$, and $Y_{Obs}$ is an 'observed' streamflow timeseries. *Y* is a set of simulated outputs that are formally

equivalent but not identical to the observation $Y_{Obs}$. Here, parameter-data pairs are simulated by a surrogate (Sect. 3.4) of ParFlow. Simulations are drawn from the same forcing scenario to limit the degrees of freedom of parameter inference.

A conditional density estimator $q_\phi(\theta|Y)$ learns the posterior density directly from simulations generated by the surrogate. $q_\phi$ is a learnable model - often a neural network - that fits to *p(θ | Y)* and can be evaluated to approximate *p(θ | Y = $Y_{Obs}$)*. (See section 3.6 for details about $q_\phi$). The procedure can be summarized as follows, after Papamakarios and Murray,

410  (2016):

1.  Propose a *prior* set of parameter vectors {$\theta$}, sampled from *p(θ)*.
2.  For each $\theta$, run the simulator to obtain the corresponding data vector, *Y*.
3.  Train the neural density estimator $q_\phi(\theta|Y)$ on the simulated set from {$\theta$, *Y*}.
4.  Evaluate $q_\phi$ at observed data vector $Y_{Obs}$ to generate a *posterior* set of parameter vectors {$\theta$} proportional to *p(θ | Y =

$Y_{Obs}$)*.

The SBI workflow and architectures used in this study are derived from a python toolbox for simulation-based inference (Tejero-Cantero et al., 2020). We direct the reader to Papamakarios and Murray (2016) for a detailed description of the structure, training, and evaluation of a neural conditional density estimator for simulation-based inference. Others (Lueckmann





et al. 2017; Greenberg, Nonnenmacher, and Macke 2019) have built on this idea to introduce MCMC-like approaches to
sequential learning of the posterior at observations to make inference more efficient. We employ a sequential learning
procedure in our workflow, as described in Appendix C.2. The hyperparameters and architectures used in SBI are shown in
Table C1.

### 3.6 Neural Conditional Density Estimators for SBI

The conditional density estimator $q_\phi(\theta|Y)$ is an essential ingredient of SBI. The neural conditional density estimator
differs from conventional neural networks (such as the LSTM) in two important ways. First, it learns a conditional probability
distribution, as opposed to a function. Second, it represents the 'inverse' model – the probability of parameters given data $p(\theta$
$| Y)$ – as opposed to the dependency of data on parameters, which is encoded in 'forward' simulators like ParFlow and its
surrogate, the LSTM. Once trained, the neural conditional density estimator is evaluated with an observation to infer a
distribution of plausible parameters, the posterior distribution $p(\theta | Y = Y_{Obs})$ (Fig. 1B).
Conditional density estimators create a model for "a flexible family of conditional densities", parameterized by a
vector of parameters ($\phi$) (Papamakarios and Murray, 2016). Density estimator parameters are not to be confused with the
simulator parameters, $\theta$. The latter are the target of inference while the former parameterize the density-estimated posterior
probability and must be learned or derived to conduct inference of simulation parameters. Deep neural networks provide new
opportunities to learn $\phi$ for complex classes of densities, which gives rise to the term *neural* conditional density estimator.
Mixture Density Networks (MDNs) are an intuitive class of conditional density estimators capable of modeling any
arbitrary conditional density (Bishop, 1994). They take the form of a mixture of $k$ (not hydraulic conductivity, K) Gaussian
components, as below.

$$q_\phi(\theta|Y) = \sum_k \alpha_k \mathcal{N}(\theta|m_k, S_k) \qquad (6)$$


where the mixing coefficients ($\alpha$), means ($m$), and covariance matrices ($S$) comprise the neural density parameterization, $\phi$.
They can be computed by a feedforward neural network.

Training an MDN is a maximum likelihood optimization problem (Bishop, 1994). Given a training set of N simulation
parameters and data pairs, $\{\theta, Y\}$, the objective is to maximize the average log probability (or minimize the negative log
probability) with respect to the parameters, $\phi$.

$$\operatorname*{argmax}_\phi \frac{1}{N} \sum_n \log q_\phi(\theta_n|Y_n) \qquad (7)$$





For a fuller description of the parameterization and training of neural density estimators, see the supplementary
material in Papamakarios and Murray (2016) or the original write-up in Bishop (1994). This study uses a specialization of this
family of neural networks called a Masked Autoencoder for Density Estimation, further described in Appendix C.1.

**3.7 Posterior Predictive Check**

A crucial diagnostic step in the SBI workflow is to check the ability of the simulator to characterize process(es) of
interest after inference has been conducted (Cranmer et al., 2020). To be more explicit, this step checks that parameters from
the inferred posterior $p(\theta \mid Y = Y_{Obs})$ can simulate streamflow data *(Y)* consistent with the observation ($Y_{Obs}$) when plugged
back into the simulator.  The simulated data should 'look similar' to the observation (Tejero-Cantero et al., 2020). Gabry et al.
(2019) describe this type of model evaluation as a 'posterior predictive check'. This predictive check is represented by the Fig.
1C.

Here, we conduct posterior predictive checks by drawing a small number of parameter sets from our inferred
parameter posterior density. In our workflow, the inferred posterior parameter density is represented by an array containing
thousands (n=5000) of plausible parameter sets. The frequency of their occurrence is 'probability weighted', in the sense that
there are very few occurrences of parameter sets in the 'tails' and many occurrences close to the mean, and improbable
parameter sets do not exist at all. For our posterior predictive check, we randomly sample (n=50) parameter sets from this
frequency-weighted parameter posterior array. We use these parameter samples to generate an ensemble of 'predicted'
streamflow time series using the LSTM.

**3.8 Calculation of Model Weights**

Bayesian Model Averaging (BMA) is a method of combining different model forms to reduce the risk of overfitting
on prediction or inference (Madigan and Raftery, 1994). The implementation explored here uses an informal likelihood
measure to assign unique probabilities, or weight, to models (both model structures and parameters) inferred by simulation-
based inference. Specifically, the sets of parameters estimated by SBI are resampled using weights based on the fit of observed
and simulated streamflow to estimate a new probability density. Given a set of *K* model structures, $M_1, M_k, ..., M_K$, this *weighted*
estimated density, $p(\theta|Y_{Obs}, w_k)$, is:

$$p(\theta|Y_{Obs}, w_k) = \sum_{k=1}^{K} p(\theta|M_k, Y_{obs}) \, w_k \qquad (8)$$


where $p(\theta|M_k, Y_{obs})$ is equivalent to the posterior parameter density, $p(\theta|Y = Y_{Obs})$, from SBI (eq. 5); and $w_k$ is the
model probability or weight, which is based on the goodness of fit of simulated data from the posterior predictive check. All
probabilities are implicitly conditional on the set of all models being considered.





In the current application, weights are calculated using the informal likelihood $L_{ik}$, a measure of acceptability for each simulation result based on its error relative to observed data. Model configurations with likelihood measures below a pre-defined limit of acceptability are rejected; the set of remaining models are assumed to be equally probable prior to weighting. Weights for each individual model configuration in the set *K* structures, each composed of a set of *I* parameter configurations, is equal to:


$$w_k = \frac{L_{ik}}{\sum_{k=1}^{K} \sum_{i=1}^{I} L_{ik}}$$     (9)

The informed reader will recognize disagreement and inconsistent usage in the literature about the likelihood function (Beven, 2012; Nearing et al, 2015). We acknowledge legitimacy in all camps, but here adopt a subjective, or informal,
likelihood as sometimes used in Generalized Likelihood Uncertainty Estimation (GLUE). We choose to use the Kling Gupta Efficiency (KGE; Gupta et al., 2009) as the likelihood metric for its utility and history rainfall-runoff model assessment. Furthermore, we note that the method is not dependent on a specific metric and others could apply this approach using a different metric if they choose.   The KGE metric is computed using the following equation:

$$KGE = 1 - \sqrt{(1-\alpha)^2 + (1-\beta)^2 + (1-\rho)^2}$$     (10)

Where α is the ratio of the standard deviation of simulated and observed streamflow data, respectively; β is the ratio of their means; and ρ is the correlation coefficient in time.

The *weighted* probability density $p(\theta|Y_{Obs}, w_k)$ is estimated using an algorithm that can broadly be explained as
sampling from a distribution, where the distribution represents the weights of each distinct parameter configuration *i* under each model structure *k*. Model indices are sampled by mapping a random target probability between 0 and 1 to the cumulative distribution of model weights. This approach can be used to sample sets of parameters from the SBI-inferred posterior parameter density weighted to high-likelihood model configurations identified by the posterior predictive check.

### 3.9 Evaluation Metrics

The performance of simulation-based inference is evaluated in terms of accuracy and precision. First, we evaluate performance with respect to the parameter posterior (the inferred parameters); and second with respect to the posterior predictive check (the ability to generate realistic data using the inferred parameters).

### 3.9.1 Evaluating the Posterior Parameter Density

Accuracy of parameter inference is evaluated using the Mahalanobis distance, $D_M(\theta_{True})$. Mahalanobis distance
measures the distance between a point and a distribution of values after Maesschalck et al. (2000), such that:





$$D_M(\theta_{True}) = \sqrt{\left(\theta_{True} - \theta_\mu\right)^T \Sigma^{-1} \left(\theta_{True} - \theta_\mu\right)} \tag{11}$$

where $\theta_{True}$ is the set of observed or 'true' parameters; $\theta_\mu$ is the mean of the posterior distribution $p(\theta \mid Y = Y_{Obs})$; and $\Sigma$ is the covariance matrix of $p(\theta \mid Y = Y_{Obs})$. In essence, Mahalanobis distance measures how far off our parameter estimate is from the 'truth'. For this study values less than two are defined as acceptable (within ~two standard deviations); this threshold was identified via trial and error.

Precision of parameter inference is evaluated in terms of the determinant of the covariance matrix of the inferred parameter posterior, $|\Sigma|$. The determinant can be interpreted geometrically as the 'volume' contained by the covariance matrix, and by extension the inferred parameter posterior distribution. Larger determinant values are less precise; smaller values more precise (4.3 Determinants and Volumes). In this study we define values less than $10^{-6}$ as acceptable, identified via trial and error.

### 3.9.2 Evaluating the Posterior Predictive Check

We evaluate the ability of the simulated ensemble of streamflow to adequately characterize the observed streamflow using the root mean squared error (RMSE) between each (n=50) simulated streamflow time series ($Y$) and the observed streamflow time series ($Y_{Obs}$). RMSE is calculated for each predication as the square root of the mean squared error, such that:

$$RMSE(Y) = \sqrt{\frac{\sum_{t=1}^{T}\left(Y_t - Y_{Obs_t}\right)^2}{T}} \tag{12}$$

where $Y_{pred_t}$ is the simulator-predicted streamflow at time $t$, taken from $Y_{pred}$; $Y_{Obs_t}$ is the observed or true streamflow at time $t$, taken from $Y_{Obs}$; T is the number of times (days) in the streamflow time series.

Accuracy of the simulator characterization of streamflow is the mean of the RMSE calculated for all n=50 $Y$ relative to $Y_{Obs}$ ($RMSE_{Ave}$). Precision of the simulator characterization of streamflow is assessed as the standard deviation of the RMSE calculated for all n=50 $Y_{pred}$ relative to $Y_{Obs}$ ($RMSE_{std}$). For both the mean and variance RMSE values less than 0.01 [scaled streamflow units], identified via trial and error, are acceptable.

### 4 Results

Here we present the outcomes of the three experiments described in Sect. 3.1. The first two experiments showcase inference problems that increase in difficulty from the easy *best case* (Sect. 4.1) to the hard *tough case* (Section 4.2). The final experiments offer workarounds by way of the *boosted case* (Sect. 4.3) and *weighted case* (Sect 4.4). The performance of the methods explored in the three experiments is first discussed in terms of one shared benchmark scenario. Then, we show the results of the three experiments on a larger shared set (n=18) of benchmark scenarios (Sect. 4.5).





### 4.1 Experiment 1 – *Best Case*

For the *Best Case* scenario, we attempt to infer the parameters of synthetic observation(s) taken from the trained surrogate simulator, such that $p(\theta \mid Y = Y_{Obs\_LSTM})$. We first infer the parameters of just one randomly selected streamflow observation, denoted with an 'A' ($Y_{Obs\_LSTM\_A}$). The set of 'benchmark' parameters ($\theta_A$) used to generate the underlying simulation are approximately 0.60 for $K_s$, and 0.85 for $M_s$. $\theta_A$ is also our benchmark in parameter space for Experiments 2 and 3.

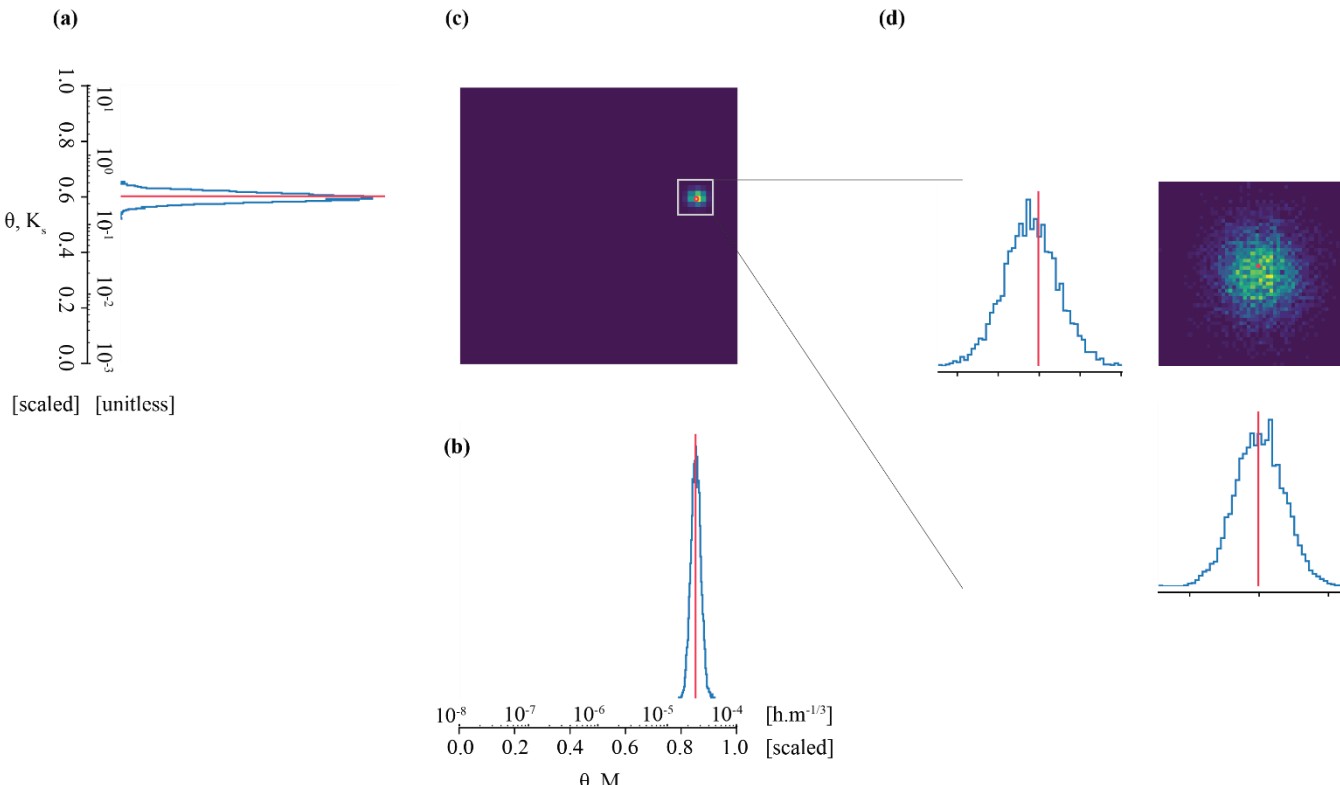

**Figure 3. The parameter posterior estimate for observation $Y_{Obs\_LSTM\_A}$ closely matches the true parameter values in the 'best' case. Subplots (a), (b) and (c) comprise a pair plot of posterior densities across the full possible parameter space; subplot (d) is zoomed in for detail. The posterior density of $M_S$ (a) and $K_S$ (b) are shown individually, and together (c). Axes are expressed in both the scale/transformed and unscaled units of the parameters. The 'true' parameters are denoted by the red line and circle, respectively.**

We accurately and precisely estimate parameters for our benchmark case (Figure 3). The pair plot approximates the posterior parameter density evaluated by the neural density estimator at the observation. In individual parameter space, narrower peaks (in blue) correspond with more confident and precise parameter estimates. In shared parameter space (c), zones of deep purple are effectively zones of no probability; zones of blue-green-yellow are zones of high probability. The benchmark parameters (i.e., the parameters used to generate the simulation) are denoted by the red line and circle, respectively. Accuracy





is evaluated by the Mahalanobis Distance, which is 3e-01; thus, the 'true' parameter set can be thought of as less than one
'standard deviation' from the central tendency of the inferred distribution. Precision is estimated by taking the determinant of
the covariance matrix. The determinant of the covariance matrix is 9e-08. This is well below our threshold of 1e-06 for
sufficiently precise parameter inference.

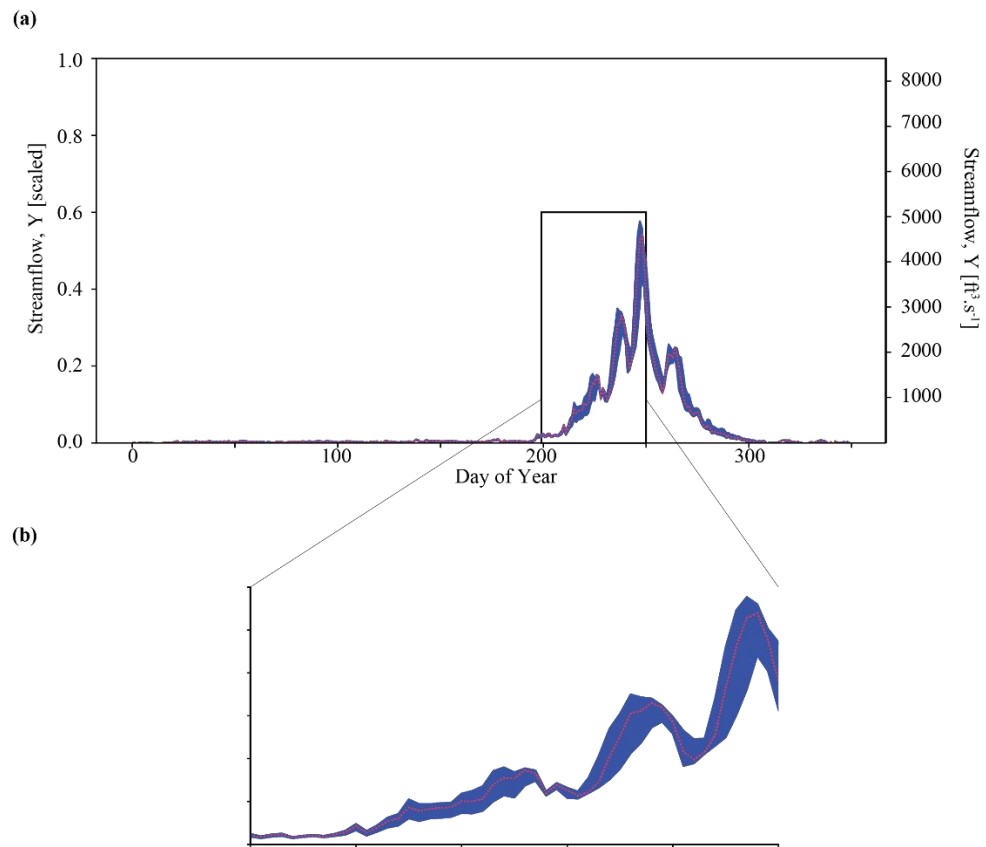

**Figure 4. Results of the posterior predictive check on synthetic observation $A_{Yobs}$ in Experiment 1 ('base' case). Subplots (a) shows streamflow simulations resulting from inference of $p(\theta|Y = A_{Yobs})$. The ensemble of predictions is bounded by blue, and observation in red. Blue lines represent time series of upper- and lower- streamflow values in this ensemble, and the red line represents the observation $Y_{Obs\_LSTM\_A}$. In subplot (b), we zoom into the area of greatest uncertainty between days 200 and 300, which correspond to the spring snow melt-off.**


Taking this one step further we can use the inferred parameter distributions to generate an ensemble of streamflow
simulations using the LSTM model and compare this to the observed streamflow (referred to as our posterior predictive check).
As show in Figure 4a, the inferred parameters generate simulation results that characterize the observed streamflow observation
reasonably well. Greater uncertainty exists around higher streamflow values over the course of the water year, as shown by
the increasing width of the uncertainty envelope after day 200 (Figure 4B). Note that this is the time of year during which



snow melt-off occurs in the Taylor River Basin. Mean and standard deviation of streamflow error are approximately 6e-03 and 4e-03 [scaled streamflow units], respectively.

### 4.1.1 Inference for many observations

In addition to conducting this analysis for one observation as described, an advantage of SBI is the low computational expense of evaluating new observations. Simulations from the process-based simulations (i.e., ParFlow) are slow and scale linearly with the number of simulations. It takes ~$10^5$ times longer to generate a ParFlow simulation (1680 seconds) than to evaluate one observation $Y_{Obs}$ using a trained neural density estimator (0.045 seconds) on a high performance computer system allocation of one CPU node with four gigabytes of working memory. Put another way, after an upfront sunk cost to learn the distributions, we can evaluate new observations, $Y_{Obs}$, practically for free. Many other techniques to parameter determination

are not 'amortized' in this way (Cranmer et al., 2020). For example, Approximate Bayesian Computation (ABC) requires restarting most steps in the inference process when new data comes available (Vrugt and Sadegh, 2013). This property of SBI can be handy in domains where the system structure (parameters) stays the same, but new observations come available all the time - as can be the case in watershed hydrology. In Appendix D, we extend Experiment 1 to evaluate the posterior parameter density for many synthetic observations ($Y_{Obs\_LSTM\_i}$).

### 4.2 Experiment 2 – Tough Case

Experiment 2 is our *tough case*. We attempt to infer the parameters of synthetic observations from ParFlow, such that $p(\theta \mid Y = Y_{Obs\_ParFlow})$. We do this using the same realization of the neural density estimator from Experiment 1 (the *best case*). The 'tough' case is a realistic test of the robustness of parameter inference. Specifically, it tests our ability to evaluate data from a different source. Unlike in the *best case,* we must deal with uncertainties related to the goodness of fit between the

simulator (the LSTM surrogate) and 'observation' (the underlying ParFlow model). We generate the posterior parameter and predictive densities to the benchmark case ($\theta_A$) explored in Experiment 1. The only difference is that $Y_{Obs\_ParFlow\_A}$ is a simulation generated by ParFlow, and not the surrogate.





**(a)**

**(b)**

Figure 5. Results of parameter inference and posterior predictive check on synthetic observation $Y_{Obs\_ParFlow\_A}$ in Experiment 2 ('tough' case). Subplots (a) and (b) show overconfident parameter inference that still results in well-constrained posterior predictive check.

Figure 5 plots the results of experiment two. Here we see that the quality of inference is somewhat degraded for the *tough case* compared to the *best case*. Parameter inference here is overconfident; it is precise but biased as indicated by the tight probability distributions and the difference between the peak probability and the observation (indicted by the red line in Figure 7A). The true parameter value does not plot in the area corresponding to highest probability. The determinant is 6e-08, which is within the same order of magnitude as the *best case*. However, the Mahalanobis Distance is much higher, at 7e0. Thus, the 'true' parameter set can be thought of heuristically as approximately seven 'standard deviations' from the central





tendency of the inferred distribution. Visual inspection of Figure 7B shows that streamflow simulations yielded by inferred parameters still characterize the synthetic streamflow observation well. However, average error is roughly twice as high for the *tough case* compared to the *best case* (1e-02 compared to 6e-03), which is approximately equal the acceptability criterion described in Sect. 3.7.

Overconfident posterior estimates are a result of the misfit between our LSTM surrogate compared to ParFlow (Figure

B1B). One interpretation of overconfident parameter inference is that the relationship between data (streamflow) and parameters ($M_s$, $K_s$) in the LSTM surrogate does not *quite* represent their relationship as it exists in ParFlow. These differences are not unexpected, because ParFlow has parameters that vary across a three-dimensional domain but are lumped together in the LSTM (See also Appendix A). This bias in the surrogate simulator increases the possibility of overconfidence in the conditional density learned by the neural density estimator. We consider this suboptimal performance in parameter inference

a consequence of 'surrogate misspecification', as described further in Sect. 6.

### 4.3 Experiment 3 – Boosted Case

A desirable approach to circumventing overconfident parameter posteriors is to make the LSTM surrogate simulator less biased. In our study, we utilize an ensemble of surrogate LSTM simulators with distinct bias stemming from surrogate misspecification subject to the initialization and selection of training data. That ensemble is then used to generate the set of

simulated pairs *{θ, Y}* to train a new neural density estimator. The underlying principle is that the overall behavior of an ensemble of surrogate simulators *in aggregate* may not be biased, even if each individual simulator has its own bias.

Experiment 3 is our *boosted* case. As in Experiment 2, we attempt to infer the parameters of synthetic observation(s) reserved from ParFlow, $p(\theta \mid Y = Y_{Obs\_ParFlow})$. As opposed to Experiments 1-2, we learn the conditional probability from an ensemble of 10 surrogate LSTM simulators instead of just one. We refer to the LSTM ensemble as a 'boosted' surrogate.

Compared to the LSTM used in Experiment 1 and 2, these LSTMs are trained for fewer epochs (100, as compared to 300) and on a smaller random split of the data (0.7, as compared to 0.6). The reserved test data is the same across the LSTMs for Experiments 1, 2, and 3. Note that we don't use an adaptive learning algorithm such as AdaBoost (Freund and Schapire, 1997), and instead we equally weight each 'weak' LSTM simulator. The neural conditional density estimator is trained by taking a random draw from the ensemble of LSTMs and using the selected LSTM to generate a forward simulation of streamflow from

a randomized parameter combination. Thousands of such draws are repeated until the conditional density has been sufficiently learned (see Appendix B for details), at which point it can be utilized for parameter inference.





**(a)**

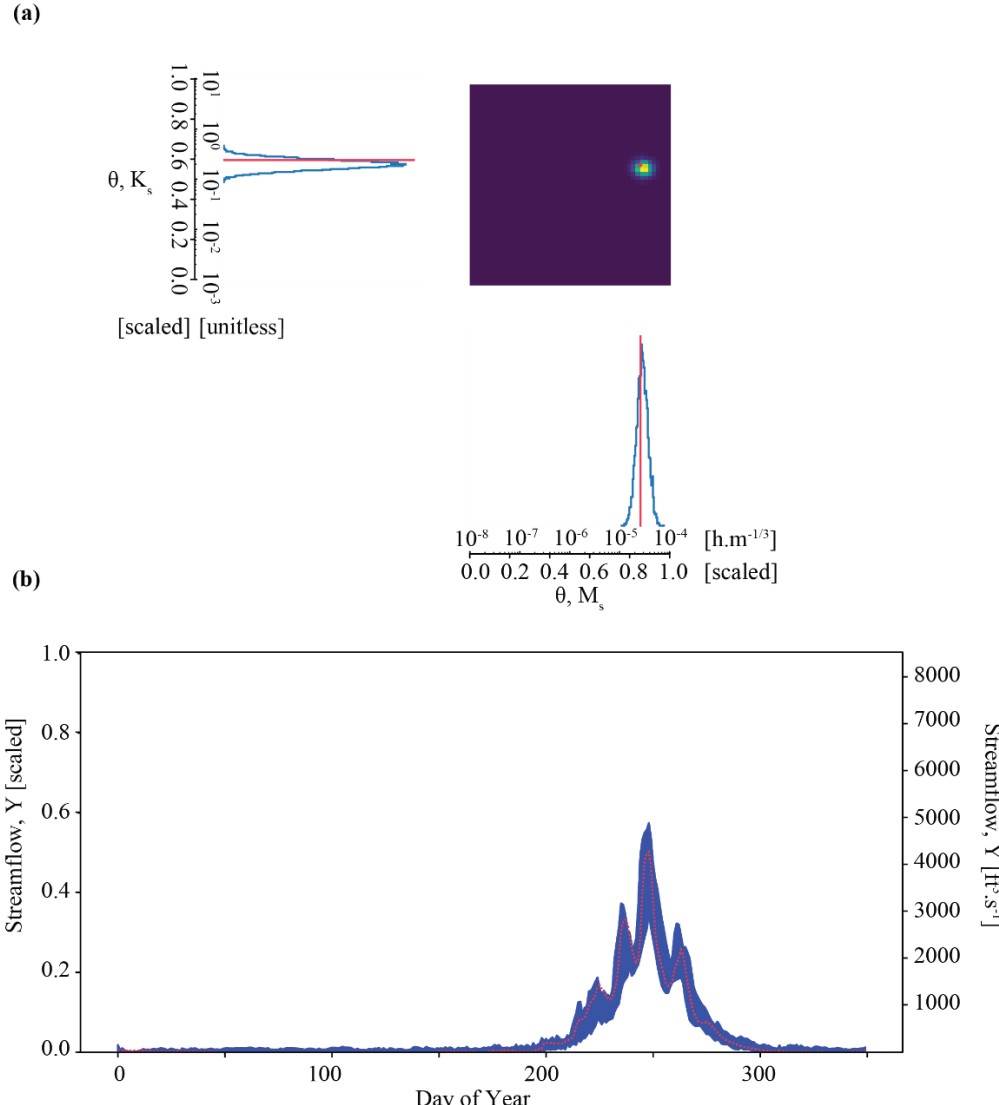

**(b)**

**Figure 6. Results of parameter inference and posterior predictive check on synthetic observation Y$_{Obs\_ParFlow\_A}$ in Experiment 3 ('boosted' case). Subplots (a) and (b) show accurate parameter inference that is somewhat less precise, resulting in a wider but still well-constrained posterior predictive check.**

Results of the *boosted case* in Experiment 3 show that we may be able to work around the issue of overconfident posteriors encountered in the *tough case* in Experiment 2. Fig. 6A shows precise and accurate parameter inference for our benchmark case in Experiment 3. The benchmark parameter values are in the area identified by the highest probability, as opposed to in Experiment 2. We note that the area of highest density is somewhat larger than in Experiment 2. The determinant is 5e-07, which is about an order of magnitude higher than the *tough case*, 6e-08. The Mahalanobis Distance is 1e0. For comparison, Mahalanobis Distance in the previous 'overconfident' experiment was 7e0. The inferred parameters generate





streamflow simulations that characterize the synthetic streamflow observation well, as shown by the posterior predictive check (Fig. 6B). We note that compared to Experiment 2 (Figure 5B) our simulations are somewhat more variable, as shown by the larger distance between the larger uncertainty envelope. The average streamflow error is about twice as high for the *boosted case* as compared to the *tough case*, (2e-02 compared to 1e-02). The standard deviation if the error is also greater (5e-03 compared to 2e-03). The sacrifice in precision with respect to both parameter inference and the posterior prediction is a consequence of using an ensemble of surrogates to simulate each parameter set.

### 4.4 Experiment 4 – *Weighted Case*

In the preceding Experiments, we aimed to rectify overconfident parameter estimates arising from SBI due to surrogate misspecification. Adding an informal likelihood measure to the inferential paradigm may help to address the issue of overconfident parameter estimates by decreasing the importance of low-credibility models. We extend the surrogate LSTM simulators from Experiment 3, each with distinct misspecification relative to ParFlow, to train a set of competing neural density estimators. Once evaluated with observed data, a metric of simulation quality representing the modeler's belief in the results of inference is used to re-weight the inferred parameter sets drawn from each of the density estimators. The added metric, the informal likelihood, emphasizes credible model structures and configurations, and safeguards against those that deviate significantly from observations.

Experiment 4 demonstrates our *weighted* case. As in Experiments 2-3, we attempt to infer the parameters of synthetic observation(s) reserved from ParFlow, $p(\theta \mid Y = Y_{Obs\_ParFlow})$. As opposed to Experiments 1-3, we use the Kling Gupta Efficiency (KGE) of the simulations resulting from the posterior predictive check as an informal likelihood measure to weight the importance of the inferred parameters. Model configurations scoring less than persistence (defined by setting next week's predicted data equal to today's observed data) are considered not credible and assigned a weight of zero. The weights, *w*, are used to condition sampling from $p(\theta \mid Y = Y_{Obs\_ParFlow})$. Weighted sampling yields a new set of inferred parameters $p(\theta \mid Y = Y_{Obs\_ParFlow}, w)$. We term this quantity the weighted posterior parameter density, an output of the methodology described in Section 3.8.

Table 3 characterizes the parameter estimates from the ensemble of competing surrogate models and density estimators for the benchmark scenario, $Y_{Obs\_ParFlow}$ and $\theta_A$. Individual ensemble members are separate rows, with the resultant *weighted* model last. Some surrogate models contain simulator configurations that are more credible than others, where credibility is represented by the average KGE of simulated data taken from the posterior predictive check for each surrogate. The average KGE (second column) for most members clusters above 0.90, and for members 7 and 9 is near a perfect match of 1.00. On the other hand, members 3 and 6 have surrogate simulator configurations below 0.80. The *weighted* KGE of 0.94 (Table 3, second column) indicates that the performance of the *weighted* model most resembles the most-credible simulator configurations, but also incorporates information from less-credible ensemble members.




**Table 3. Calculation of the weighted model from surrogate models for baseline synthetic observation $Y_{Obs\_ParFlow\_A}$.**

| Model[1] | KGE[2] | Cumulative Weight (%)[3] | Rejections (%)[4] | $D_M$[5] | $|\Sigma|$[5] |
|---|---|---|---|---|---|
| 9 | 0.97 | 13.5% | < 0.200 % | 3.8 | 2.90E-07 |
| 7 | 0.97 | 13.4% | < 0.200 % | 0.3 | 7.20E-08 |
| 5 | 0.96 | 13.3% | < 0.200 % | 2.3 | 1.40E-07 |
| 4 | 0.96 | 13.2% | < 0.200 % | 5.4 | 1.20E-07 |
| 8 | 0.95 | 13.1% | < 0.200 % | 4.6 | 1.20E-07 |
| 2 | 0.90 | 12.4% | < 0.200 % | 3.8 | 1.30E-07 |
| 0 | 0.86 | 11.7% | 2.20% | 1.7 | 1.20E-07 |
| 1 | 0.85 | 9.34% | 23.0% | 7.0 | 7.50E-07 |
| 3 | 0.78 | 0.045% | 99.6% | 4.5 | 1.70E-07 |
| 6 | 0.77 | < .00100% | 100.0% | 6.6 | 1.80E-07 |
| *Weighted*[6] | *0.94* | -- | -- | *1.1* | *3.00E-06* |

1.  Members of the ensemble of surrogate models, and their associated neural density estimators (n=9).
2.  Average Kling Gupta Efficiency (KGE) calculated from unweighted posterior predictions.
3.  Each posterior predictive simulation is weighted by the associated KGE; simulation weights are zero where poorer than persistence (KGE<0.81). The value in this column is the sum of the individual weights of 5000 predictive simulations taken for each surrogate model.
4.  Count of rejected (zero weight) simulator configurations divided by the total number of configurations for each model ensemble member.
5.  Mahalanobis Distance, $D_M$, and determinant, $|\Sigma|$, calculated by comparing are $\theta$, $M_s = 0.85$ and $\theta_A$ $K_s = 0.60$ to the unweighted parameter posterior $p(\theta \mid Y = Y_{Obs\_ParFlow\_A})$ for each surrogate model
6.  The weighted posterior parameter density $p(\theta \mid Y = YObs\_ParFlow, w)$, derived by resampling the posterior densities using individual weights.

The ensemble members with many credible configurations have a higher weight, or importance, in the *weighted* model. The weights, which are calculated from the sum of KGEs of the simulator configurations, are presented in the third column as the cumulative weights. Because predictive checks from members 8, 4, 5, 7, and 9 contain an equivalent number of credible simulator configurations, they are nearly equally weighted. Less importance assigned to some members in the *weighted* model is derived from lower likelihoods and rejection (fourth column) where KGE is less than the limit of acceptability (i.e. <0.81). Surrogates 1, 3, and 6 have many rejected configurations, which are assigned a weight of zero.

The *weighted* model is considered more accurate than all but one of the ensemble members. The relative accuracy of parameter estimates is presented in the fifth column as the Mahalanobis Distance, $D_M$, of the posterior parameter density for each surrogate. This increase in accuracy reflects in part that higher-weighted members are associated with more-accurate parameter estimates compared to those that are lower-weighted. Note that the weighted parameter estimate is also less precise compared to that of the individual surrogates, as represented by the determinant $|\Sigma|$ in column six.



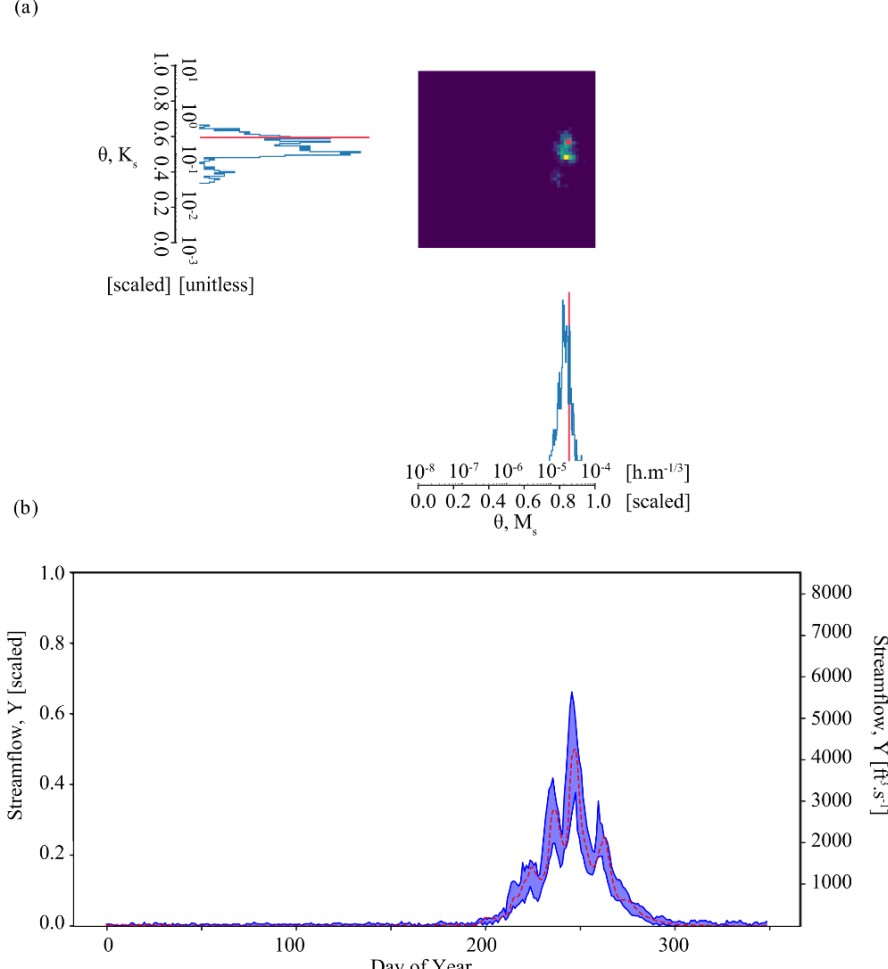

**Figure 7. Results of parameter inference and posterior predictive check on synthetic observation in Experiment 4 ('weighted' case). Subplot (a) shows accurate parameter inference that is somewhat less precise and discontinuous, focused on model structures and parameter combinations that are defined by a higher likelihood. The result is a narrow, well-constrained posterior predictive check in (b).**

Results of the *weighted case* in Experiment 4 demonstrate that it is a viable approach to the issue of overconfident posteriors encountered in the *tough case* in Experiment 2. Fig. 7A shows accurate parameter inference for our benchmark case in Experiment 4. As in Experiment 3, the benchmark parameter values are in the area identified by the highest probability. The Mahalanobis Distance, 1.1, is like that of Experiment 3. The geometry of the area of the highest density differs from Experiment 3, covering a larger area due to differences in the unweighted parameter estimates associated with each surrogate. As a result, the parameter estimate is less precise: the determinant $|\Sigma|$ is 3e-06, which is about an order of magnitude higher than the *boosted case*, 5e-07. The inferred parameters generate streamflow simulations that characterize the synthetic





streamflow observation well, as shown by the posterior predictive check (Fig. 7B). We note that compared to Experiment 3 (Figure 6B) our simulations are about as variable. The average streamflow error is similar for the *boosted case* as compared to the *weighted case* (2e-02). The standard deviation of the error is also very similar (5e-03 compared to 6e-03).

## 4.5 Summary of Experiments 1-4

Previously, we compared the performance of simulation-based inference in Experiments 1 (*best case*), 2 (*tough case*), 3 (*boosted case*), and 4 (*weighted case*) on only one benchmark parameter set. In this section, we expand the comparison of SBI across the experiments to a larger number (n=18) of parameter sets and corresponding observations. In the case of Experiments 1 and 2, the same neural density estimator was utilized to conduct inference. For Experiment 3, an ensemble approach was used to create one new neural density estimator; for Experiment 4, likelihood-weighted parameter estimates

from an ensemble of neural density estimators was used. In the case of Experiments 2- 4, the mock data are the same benchmark streamflow simulations from ParFlow; for Experiment 1, the observations are taken from the surrogate. All four experiments utilize mock data corresponding to the same test parameter sets, to make an apples-to-apples comparison. For reference, those test parameter sets are plotted relative to parameter space in the Fig. B1A. The results of the analysis of multiple (n=18) parameter sets are shown by the box plots in Fig. 8.

### 4.5.1 The precision and accuracy of parameter inference

In general, the parameter estimates from the four experiments are accurate and precise, as shown in Fig. 8A and 8B. The *best case* (Experiment 1) tends to be both precise and accurate. Compared to Experiment 1, the *tough case* (Experiment 2) tends to be just as precise but less accurate, while the *boosted case*. This is to be expected as we made the problem harder for Experiments 2-4 by not assuming a perfect surrogate.  Experiment 3 tends to be less precise but more accurate than

Experiment 2. Compared to Experiment 3, the *weighted case* (Experiment 4) tends to be yet less precise and more accurate. A couple of second-order discussion points arise from Figs. 8A and 8B.

The resulting box plots of the determinant, a metric for the precision of inference, are shown in Fig. 8B. Here we see that the training of the conditional density estimator – and not the source of the observations – seems to define the precision of inference. The box plots show parameter inference is more precise (i.e., the determinant smaller) for Experiments 1 and 2,

compared to Experiments 3 and 4. Experiments 1 and 2 use synthetic observations from different sources (the LSTM surrogate and ParFlow, respectively), however they are both evaluated using the same neural conditional density estimator; note the similar behavior of the determinant in the first two experiments. On the other hand, the determinant behaves quite differently in Experiment 2 compared to Experiments 3 and 4; all three experiments use synthetic observations from ParFlow, but use different configurations of the neural conditional density estimator. In the case of Experiment 3 (the *boosted case*), differences

within an ensemble of LSTM surrogates are lumped into the training of one neural density estimator; in the case of Experiment 4 (the *weighted case*), those differences are incorporated in the training of separate neural density estimators. Results show that Experiment 3 is associated with greater precision in parameter inference (i.e. smaller determinant) compared to Experient





4, as shown by the expanded volume of the parameter estimates in Figs. 7A compared to 6A. The lumping approach in the *boosted case* may smooth differences between the surrogates, de-emphasizing parameter combinations in the tails of the

separated posterior densities used in the *weighted case*. The likelihood-weighting and limits of acceptability also influence the distribution of the parameter estimate, but not in a manner that significantly decreases its precision. More fundamentally, the precision of parameter inference for those methods seems to reflect the simulator(s) (i.e., the variety in simulated responses, $Y$, to parameter configurations, $\theta$), and not contain much, if any, information about the goodness-of-fit between observations, $Y_{obs.}$ and simulated data, $Y.$[3]

Box plots of the Mahalanobis[4] distance, a metric of the accuracy of inference, are shown in Fig. 8A. The box plots show that parameter inference in Experiments 2 and 3 degrade in accuracy compared to Experiment 1, while parameter inference from Experiment 4 is nearly as accurate. The box plots also demonstrate that parameter inference is in general more accurate for the *boosted case* (Experiment 3) compared to the *tough case* (Experiment 2). However, the Mahalanobis distance is greater at some outlier points in the *boosted case* (Figure 7B). What this means is that while the *boosted case* yields more

accurate inference in some parts of parameter space (for example, the benchmark parameter set $\theta_A$ explored throughout the earlier results sections), this implementation is no silver bullet for averting overconfident parameter estimates. On the other hand, the *weighted case* introduced in Experiment 4 is consistently associated with much smaller Mahalanobis distances compared to either the *tough* or *boosted* cases. The apparent accuracy of the *weighted case* can be attributed to the likelihood-based weighting and limits of acceptability methodology, as well as the decrease in precision due to drawing from a set of

competing density estimates.

---

[3] This behavior is also observed in Figure D1A, which shows that the determinant exhibits a fixed pattern across parameter space.

[4] Note that Mahalnobis distance is a precision-weighted metric of distance, unlike Euclidean distance. These numbers should not be considered raw distance.



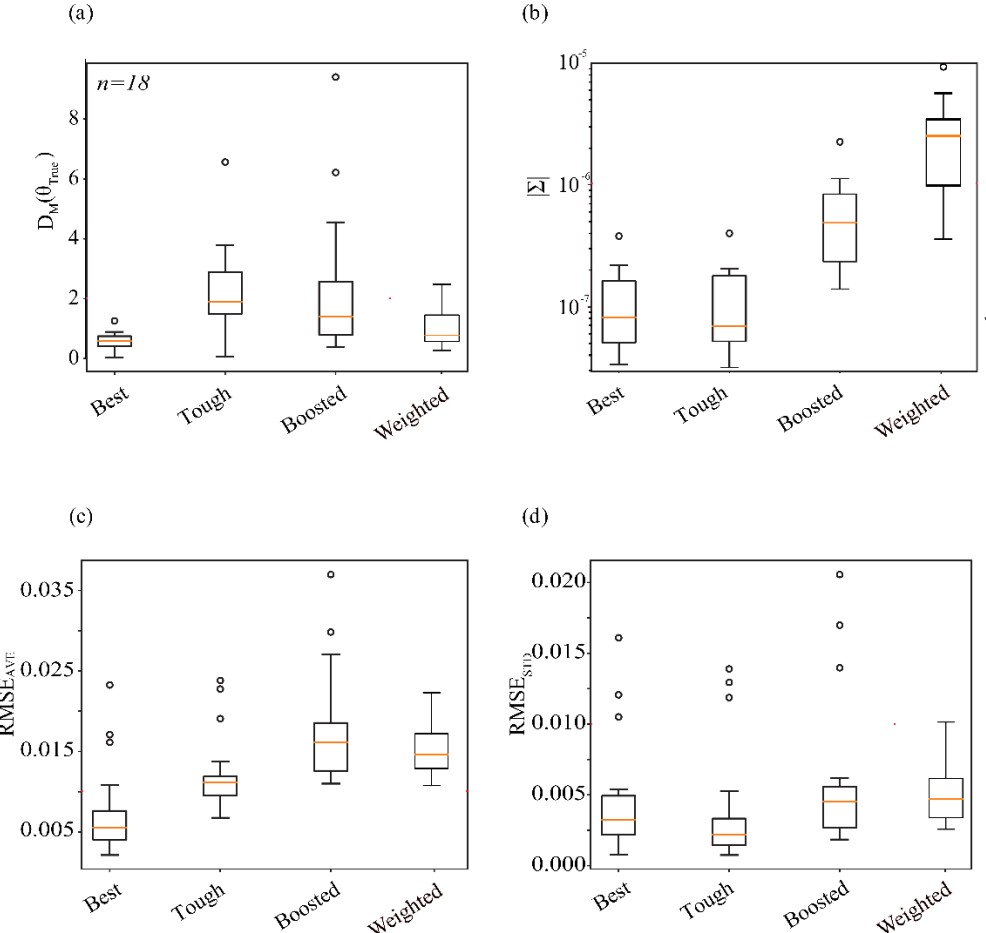

**Figure 8: Comparative plots showing the performance of simulation-based inference of parameters and predicted quantities across a set of n=18 test data. We compare the results of Experiments 1 ('base' case), 2 ('tough' case), 3 ('boosted' case), and 4 ('weighted' case). Subplots (a) and (b) show the accuracy and precision of parameter inference. Accuracy is shown in subplot (a) via the Mahalanobis Distance of the posterior parameter density. Precision is shown in subplot (b) via the Determinant, |Σ|. Subplots (c) and (d) show the accuracy and precision of the posterior predictive check. Subplot (c) shows the average of the error, RMSE_Ave of streamflow ensembles relative to 'truth', which can be thought of as a measure of accuracy. Subplot (d) shows the standard deviation of the error, RMSE_std of streamflow ensembles, which can be thought of as a measure of precision. Values closer to the x-axis are more desirable.**

### 4.5.2 The precision and accuracy of posterior predictions

Taking this one step further we can use the inferred parameter distributions to generate an ensemble of streamflow simulations using the LSTM and compare this to the observed streamflow (referred to as our posterior predictive check). As shown in Fig. 8C and 8D, the posterior predictions are precise, and generally fairly accurate. Fig. 8C shows the average of the error (RMSE_Ave) between the simulated streamflow timeseries and the observed time series, with lower average error corresponding to greater accuracy. Streamflow prediction accuracy decreases between Experiments 1, 2, and 3. This is represented by the fact that the RMSE_AVE increases nearly 3-fold across each of our experiments (median ~0.005 in *best case*,





~0.010 in *best case*, and ~0.015 in *boosted case* [scaled streamflow units]). The degradation in posterior predictive accuracy is related to degradation in the accuracy of parameter inference (Figure 8A). Fig. 8D shows the variability of the error (RMSE$_{STD}$) between the simulated streamflow timeseries and the observed time series, with lower error variability corresponding to greater precision. We see that the central tendency of the RMSE$_{STD}$ of streamflow simulations for the *base, tough,* and *boosted* cases are all similar. Streamflow posterior predictions across all three experiments remained precise, in spite of the breakdown in the accuracy.

In Experiment 4 (the *weighted case*), the posterior predictive accuracy (RMSE$_{AVE}$) and the average variability (RMSE$_{STD}$) is improved compared to Experiment 3. Improvement is seen in the outliers, where simulator configurations with a poor fit relative to observed data are assigned low or no weight in Experiment 4 based on the informal likelihood. Importantly, KGE was used in the calculation of the informal likelihood. So, conclusions about the accuracy and precision of posterior predictions associated with the four Experiments may differ as measured by KGE as opposed to RMSE.

The multi-observation comparison helps us to generalize some insights. 1. Inference results are often desirable; in particular, SBI seems to result in precise parameter inference across all conditions. 2. Parameter inference with a well-trained surrogate simulator is precise, but not always suited for conducting inference on observations with an uncertain relationship to simulated data (as in Experiment 2). 3. The performance of posterior predictive checks is dependent on both the performance of the simulator and the neural density estimator. As such it can be a valuable tool in assessing the performance of parameter inference. 4. Although a density estimate derived from an ensemble of simulators (as in Experiment 3) may yield more accurate parameter inference, overconfident parameter estimates are a recalcitrant problem for some observed data. 5. In Experiment 4, an approach to likelihood-weighting parameter estimates from SBI was demonstrated to overcome the problem of overconfidence in these controlled experiments.

## 5 Discussion

As users of hydrologic tools such as high-fidelity, process-based simulators, we are often interested in finding the model configuration(s) most consistent with watershed observations and established physical theory. In practice, this gives rise to uncertainty about whether a model is "adequate", as measured by its predictive ability and structural interpretability (Gupta et al, 2012). In the special case where a correct model structure exists, the modeler's task is to conduct a specification search (Leamer, 1978) to identify it; other candidate models inconsistent with observations and theory can be said to be "misspecified" (Cranmer et al., 2020). One example of misspecification in this work is underscored by the misfit between the process based ParFlow and the surrogate LSTM simulators. We call this special situation surrogate misspecification.

Our research shows that using a misspecified surrogate to conduct simulation-based inference for a process-based hydrologic simulator can yield erroneous parameter estimates. These 'overconfident' estimates occur because the neural density estimator learns the conditional relationship between parameters and data only from the surrogate simulator. Thus, SBI explicitly infers inputs to the surrogate and *not* parameters of the process-based simulator. Given surrogate misspecification,





the inferred values of parameters may not retain their physical significance to the process-based simulator; this can be a barrier to the interpretability of those models identified by inference.

We demonstrate that erroneous parameter estimates due to surrogate misspecification can be addressed through informal Bayesian model averaging (BMA). This approach to BMA applies a performance check – the informal likelihood – to weight and reject models identified by SBI. Notably, the likelihood and related limits of acceptability are chosen by the
practitioner based on modelling goals. Thus, broadly, informal BMA belongs to the class of approaches to encode expert / domain knowledge into a deep learning framework (e.g Reichstein et al., 2019). More specifically, SBI conducts a preliminary search of parameter space for plausible model structures and configurations, and the likelihood test incorporates expert-defined information about model adequacy into the parameter estimate. Overconfident parameter estimates carry the risk of under-representing the uncertainty of the inferences we draw form models. Our work shows that, with these two methods in
combination, erroneously overconfident parameter estimates are less likely to occur than in standalone SBI.

In our experiments we focused investigation on SBI and not the process based model.  Extending this methodology to observed data requires consideration of many additional sources of uncertainty compared to the synthetic case. Among these is much deeper uncertainty about which model structure(s) is (are) appropriate. In the synthetic experiments presented, the relationship between the model (the surrogate) and the data-generating process (ParFlow) is well-defined; the surrogate is
learned directly from ParFlow. Yet for real hydrologic problems, physics-based models are nearly always simplified representations of real data-generating processes; stumbling upon a "true" representation is unlikely, even impossible. Moreover, physical parameters like hydraulic conductivity (K) and Manning's roughness (M) are themselves conceptual quantities and are almost never known at the scale we care about, making estimates difficult to validate (Oreskes et al., 1994). In this real-world case, the modeler's search may be for a set of adequate model structures and configurations (i.e. Gupta et.
al, 2012), where adequacy is subjectively defined. Here, a reasonably good estimate of the hydrologic variable (i.e., streamflow) is often what watershed scientists strive for (Van Fraassen and others, 1980).  For completeness, a worked example demonstrating the estimation of parameters using the current model formulation and observed streamflow data from the Taylor basin is presented in Appendix E.

The critic might suggest that not enough was done tailor the present analysis to real world data. We disagree on the
grounds that our purpose here is to rigorously present and evaluate a method for parameter inference given well-defined constraints. The challenge of this goal is real and relevant. In fact, this work seems to show an upper bound for the performance of SBI where undiagnosed structural error exists. A novel model averaging approach inspired by Approximate Bayesian Averaging (BMA) and General Likelihood Uncertainty Estimation (GLUE) (Hoeting, 1999; Beven and Binley, 1992) is demonstrated to be an important check to SBI, in presented synthetic and real examples. Further comparison to observations
would instead shift the focus of this work from the quality of the SBI and BMA methods to the quality of the underlying hydrologic simulator. Logical next steps to further extending this methodology to the real case are outlined below.

Adding additional complexity to the training set for the surrogate simulator (i.e., exploring a larger number of parameters configurations, their spatial variability, or multiple forcing scenarios) may help yield better parameter estimates





and associated predictions. Many of the practitioners of simulation-based inference advocate packing as much complexity into

models as possible (Alsing and Wandelt, 2019). High-resolution process-based simulators (such as ParFlow) can be used to explore the real-like behaviors of watersheds across a great number of variable and parameter configurations by leveraging deep-learned surrogates and SBI. Beyond the informal BMA evaluation of SBI presented here, it may also be important to control for the tradeoff between complexity and parsimony in this expanded set of model structures and configurations. This could be achieved using a framework similar to the Akaike Information Criterion (e.g. Schoups et al., 2008), which adds a

penalty term related to the number of estimated physical parameters in the likelihood evaluation. A similar 'penalty for complexity' concept was explored in traditional applications of Bayesian Model Averaging for linear regression models through Occam's Window (Madigan and Raftery, 1994).

Including additional watershed observation types (i.e., groundwater, soil moisture) in the inference workflow could also improve estimates of the physical parameters for real systems, and the predictions associated with complex simulators.

However, observations in hydrology – particularly about groundwater systems – are generally sparse. This presents a problem. One option is to observe that complexity *better*. New spatially distributed 'big data' products that leverage remote sensing to offer new opportunities to observe hydrologic variables like soil moisture (Mohanty et al., 2017; Petropoulos et al., 2015). The extension of the methodology to real-world observations will also need to consider the role of data quality, adequacy (Gupta et al., 2012), and disinformation (Beven and Westerberg, 2011) and the challenge of defining limits of acceptability regarding

model performance.

## 6 Conclusion

Our investigation implements simulation-based inference (SBI) to determine parameters for a spatially distributed, process-based watershed simulator. We believe this research is among the first to apply contemporary SBI to watershed modeling. The implementation employed here has a couple of noteworthy features:

a. We use deep learning to train a surrogate Long Short-Term Memory (LSTM) on the original physically based simulations (from ParFlow). This allows for quick and comprehensive exploration of simulation results for which we have corresponding observations, such as streamflow at a basin outflow in a watershed.

b. A density-based neural network leverages the capacity of the surrogate to generate simulations quickly to learn a representation of the full conditional density, $p(\theta|Y)$, of parameters given data. This learned conditional density can

be evaluated using observations to determine the parameter posterior density, $p(\theta|Y = Y_{Obs})$. This parameter posterior represents our 'best guess' of what the parameters for our simulator should be.

We demonstrate that this approach to SBI can generate reasonable estimates of the parameters of a hydrologic simulator, ParFlow, through a set of synthetic experiments. We show in Experiment 1 (the *best case*) that SBI works well in controlled settings in which we assume that our surrogate LSTM simulator is accurate. Moreover, this experiment highlights

how, once learned, the model of the conditional density can be used to determine the process-based parameters rapidly and





effectively for many observations without the need for additional process-based simulations. That's particularly valuable when simulations are costly, as is often the case with high-resolution, transient simulators used in the field of watershed modeling.

We show in Experiment 2 (the *tough case*) that SBI produces a set of probable parameters with precision in settings where the simulator does not represent the underlying system generating the observation perfectly. These inferred parameters

are used to generate reasonable streamflow simulations relative to observations. However, the *tough case* shows that parameter inference is not always accurate with respect to the physics-based simulator that was used to train the surrogate. This undesirable characteristic (of precision but not accuracy, or 'overconfidence') arises from issues related to the structural adequacy of the simulator, which is well-recognized in the literature as an impediment for accurate parameter inference (Cranmer, 2020). The controlled nature of Experiment 2 explores the special case of 'surrogate misspecification'. This special

case arises from a mismatch between the surrogate and the process-based simulations from ParFlow. In inference, surrogate misspecification gives rise to error in estimates of the physical parameters. We show that sources of this error can be quite difficult to diagnose, although conducting a posterior predictive check is a qualitative way of ascertaining the extent of simulator bias.

In Experiments 3 and 4 (the *boosted* and *weighted cases*, respectively), we attempt to address the issue of

'overconfident' parameter inference due to misspecification. In Experiment 3, we use an ensemble of 'weak' surrogate simulators (instead of just one 'strong' surrogate simulator) to learn the full conditional density. The underlying principle is that the behavior of an ensemble of surrogate simulators *in aggregate* may not be biased, even if each individual simulator has its own bias. This may 'wash out' the negative effects of surrogate misspecification on parameter inference. Evidence from the *boosted case* shows this approach reduces the occurrence of overconfident parameter estimates, but is not a silver bullet

for conducting accurate inference.

In Experiment 4 (the *weighted case*), the modeler assigns a "measure of belief" to parameter estimates from a set of competing conditional density models, reflecting their confidence in its validity. This measure of belief – or informal likelihood (i.e. Beven and Binley, 1992) – is used to weight and reject models identified by SBI. The underlying principle is that SBI conducts a preliminary search of parameter space for plausible model structures and configurations, and the likelihood test

incorporates expert-defined information about model adequacy into the parameter estimate. The *weighted case* is demonstrated to solve the problem of overconfident parameter estimates introduced by surrogate misspecification.

The results of Experiments 2, 3, and 4 demonstrate progress towards being able to implement SBI in hydrological domains subject to uncertainty we can benchmark (i.e., the misspecification of the surrogate). Additional work is needed to address deeper uncertainty about the structural adequacy of the underlying physics-based model. This uncertainty often exists

in watershed modeling – due to (e.g.) natural heterogeneities in the subsurface, approximations in process parameterizations, and bias in the meteorological input data – that can seldom be fully 'accounted for'. The notion of structural 'adequacy' is thus nearly always subjective (Gupta et. al, 2012). In many 'real world' applications, a calibrated estimate of the hydrologic variable (i.e., streamflow) is what watershed scientists strive for. Enhancing standalone SBI with the likelihood-weighting methodology introduced in Experiment 4 embraces this principle of subjective 'adequacy' and is broadly extendable to more complex





inference problems in watershed modeling. Where no models are identified as adequate, an obvious next step is to expand the

simulator to explore more and different configurations of parameters and input variables.

**Appendix A The Process-Based Simulations (ParFlow)**

**Table A1: The relationship between ParFlow and LSTM static inputs (e.g., parameters, θ), dynamic inputs (e.g., meteorological forcings, X), and dynamic outputs (e.g. streamflow, Y). ParFlow variables must be 'compressed' into lower-dimensional**
**representations in order to be used in the LSTM.**

|  | **ParFlow Description** | **LSTM Description** |
|---|---|---|
| **Parameters, θ** | a) 2-dimensional homogeneous Manning's Roughness, $M$ <br> b) 3-dimensional heterogeneous Hydraulic Conductivity, $K$ <br> *(Other static inputs, such as soil properties and land cover, are not used by LSTM)* | a) Scalar value, $M_s$, set for all values of $M$ <br> b) Scalar factor, $K_s$, multiplied by all values of $K$ <br> *(Both are log transformed and re-normalized to be between 0 and 1)* |
| **Dynamic Outputs, Y** | Hourly, 3D spatially distributed pressure field | Daily, 1-dimensional discharge time series (length=350) at i,j location corresponding to USGS gage 09110000, as follows: <br> 1. *Gridded discharge calculated using surface pressure, slopes, Manning's, resolution via the overland flow equation for each hourly time step (n=8,760) of one year of ParFlow results* <br> 2. *Slice at i,j location and calculate daily average* <br> 3. *Remove first 15 days of record (burn in time), and renormalize values between 0 and 1* |
| **Dynamic Inputs, X** | Hourly, 2D spatially distributed meteorological forcings, including: <br> • *DLWR: Long Wave Radiation [W.m-2]* <br> • *DSWR: Short Wave Radiation [W.m-2]* <br> • *Press: Atmospheric pressure [pa]* <br> • *APCP: Precipitation [mm.s-1]* <br> • *Temp: Air Temperature [K]* <br> • *SPFH: Specific humidity [kg.kg-1]* <br> • *UGRD: East-west wind speed [m.s-1]* <br> • *VGRD: South-to-North wind speed [m.s-1]* | Daily, 1D time series (length=350) for each (n=8) forcing: <br><br> *(Except for APCP, forcings are averages taken over space and time for all hours (n=24) in each day. APCP is the sum over space and time for all hours (n=24) of precipitation each day.)* |





**Table A2: ParFlow was run many times under different parameter configurations. This table shows the scalar factors used to modify spatially distributed Manning's Coefficient and Hydraulic Conductivity. We call these factors $K_s$ and $M_s$, respectively, to keep the distinction between them and ParFlow's parameters clear.**

|  | $K_s$ (Scaling factor times whole domain)[unitless] | $M_s$ (Constant across domain), [h/m^(1/3)] |
|---|---|---|
| **Scalar Parameters** | 0.001, 0.01, 0.025, 0.05, 0.075, 0.1, 0.25, 0.5, 0.75, 1, 2.5, 5, 7.5, 10 | 1e-8, 1e-7, 2.5e-7, 5e-7, 7.5e-7, 1e-6, 2.5e-6, 5e-6, 7.5e-6, 1e-5, 2.5e-5, 5e-5, 1e-4 |


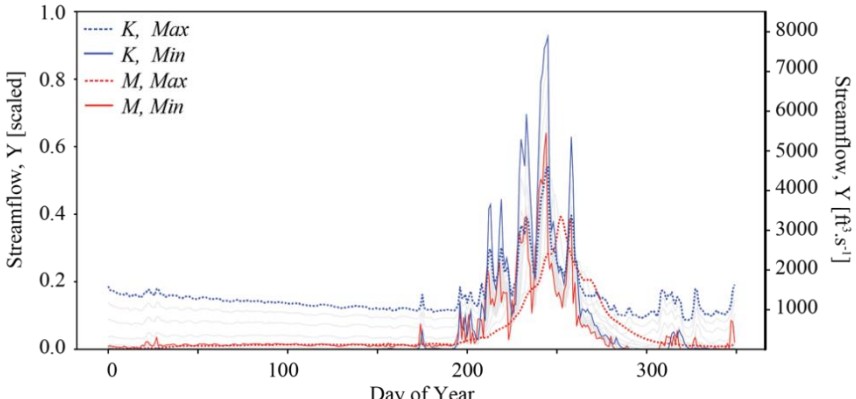

**Figure A1: Sensitivity of ParFlow-generated streamflow time series for water year 1995 to perturbations of Hydraulic Conductivity and Mannings. We show sensitivity holding each of $K_s$ and $M_s$ constant at 0.1 and 5e-6, respectively, while varying the other across the range of parameters explored in Table A2.**

**Appendix B The Surrogate Simulator (LSTM)**

**Table B1: Relevant notes on architecture, training, and hyperparameters for the surrogate LSTM simulator.**

|  | LSTM | Further Description |
|---|---|---|
| **Number of Epochs** | 300 | Number of times iterating through training loops |
| **Batch Size** | 50 | Batching during training |
| **Input Size** | 10 | Number of input features |
| **Hidden Layers** | 1 | Number of hidden layers |





| Hidden Size | 10 | Number of hidden nodes / layers |
|---|---|---|
| Number of Classes | 1 | Number of nodes in output |
| Objective Function | MSE | Mean Squared Error |
| Optimizer | Adam | |
| Learning Rate | 0.001 | |
| Train-Validation-Test Split | 0.7, 0.2, 0.1 | Simulations were divided into sets based on their parameters, such that each member characterizes the streamflow response (encoded as a year-long timeseries) to an individual pair of parameter values $K_s$ and $M_s$. We conduct the train-validation-test split in a pseudo-Latin hypercube manner across parameters space. |

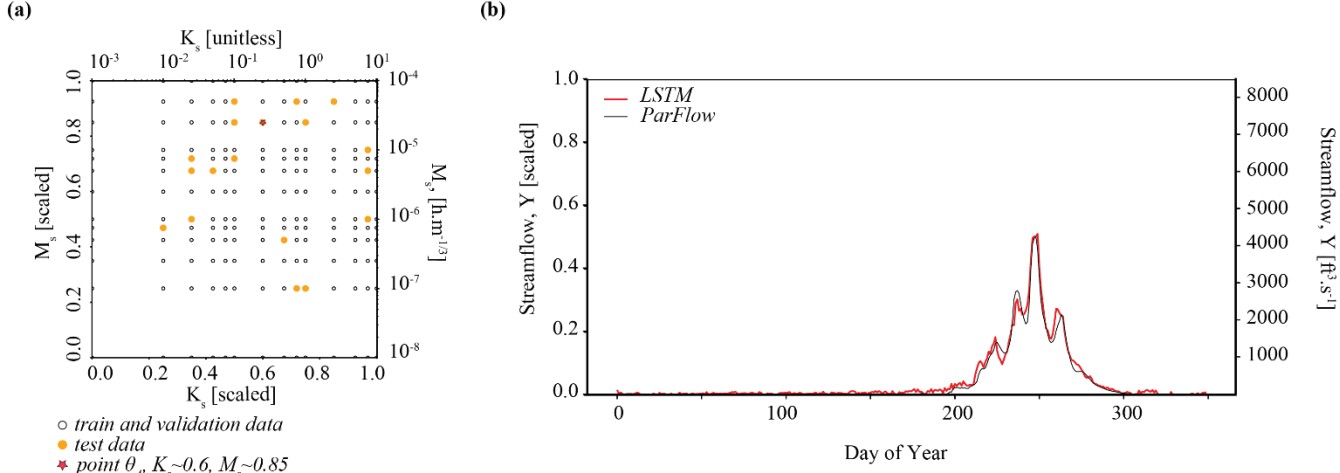

**Figure B1: Plots show the train/validation and test split for the LSTM surrogate trained on n=183 ParFlow simulations. In (a), the**
**locations in parameter space where ParFlow simulations were run. The surrogate is trained and tested at orange dots. In (b), a**
**comparison of ParFlow to LSTM streamflow simulation generated at benchmark parameter set $\theta_A$ $K_s$~0.6, $M_s$~0.85. The fit between**
**ParFlow and LSTM is explored more in the results.**

## Appendix C Improved Components for SBI

Deriving implicit statistical models using density estimation techniques is not new (Diggle and Gratton, 1984).
However, these traditional approaches suffer from some shortcomings, including sample efficiency and inference quality, as





described further in Cranmer, Brehmer, and Louppe 2020. We show two components of the density based SBI workflow utilized here that have benefited due to recent innovations: Masked Autoencoders for Density Estimation (MADEs) and sequential neural posterior sampling.

## C.1 Masked Autoencoder for Density Estimation (MADE)

945        While mixture density networks have a long operational history, there have been more recent innovations in using neural networks to learn and represent conditional probability distributions. This study utilizes a class of neural density estimators called Masked Autoregressive Flows (Alsing et al., 2019), which share some of the underlying principles described for Mixture Density Networks. Masked Autoregressive Flows arise from the principle that "any probability density can be factorized as a product of one-dimensional conditionals" via the chain rule (Alsing et al., 2019); these one-dimensional
conditionals are parameterized by a fully connected neural network known as a Masked Autoencoder for Density Estimation (MADE) (Uria et al., 2016). Masked Autoregressive Flows are composed of 'stacks' of Masked Autoencoder for Density Estimations, to add flexibility (Papamakarios et al., 2018) . A detailed description of these methods is beyond the scope of this paper.

## C.2 Sequential Neural Posterior Estimation

955        We use a sampling technique called Sequential Neural Posterior Estimation (SNPE) to speed up and improve the evaluation of a trained neural conditional density estimator. By evaluation, we here mean using data $Y$ (most typically observed data, $Y_{Obs}$) to generate a posterior estimate $p(\theta \mid Y = Y_{Obs})$ (step 4 in Sect. 3.5). The need for SNPE arises from the challenge that drawing simulation parameters from the full prior distribution is wasteful (Papamakarios et al., 2018; Lueckmann et al., 2017; Greenberg et al., 2019). This is due to the fact that data simulated from some parts of parameter space have higher or
lower posterior density for $Y_{Obs}$. SNPE iteratively refines the posterior estimate to make inference more efficient and flexible, as described by Greenberg et al, 2019.

        Details related to the architectures, hyperparameters, training, and evaluation of neural density estimators are shown in Table C1. Decisions about hyperparameters were made via trial and error. It's important to note that the goal of our work is not to create the most robust neural density estimator model, but to explore inference under a variety of different conditions.


**Table C1: Hyperparameters and model architecture for neural density estimation. See also (Tejero-Cantero et al., 2020).**

| Hyper- parameter | Value | Significance |
|---|---|---|
| **Inference Method** | SNPE_C | Sequential Neural Posterior Estimator (see text) |
| **Neural Density Model,** $q_\phi(\theta\|Y)$ | MAF | Masked Autoregressive Flow (see text) |





| Hidden Features | 10 | number of hidden layers in each MADE of $q_\phi(\theta|Y)$ |
|---|---|---|
| Number of Transforms | 2 | Number of flows (transforms) between MADEs in $q_\phi(\theta|Y)$, MAF |
| Prior_min, Prior_max | 0.0, 1.0 | Minimum and Maximum possible values of $q_\phi(\theta|Y)$, $K_s$ and $M_s$ |
| Prior Function | Uniform | All values *a priori* equally possible in parameter space |
| Number of simulations | 1000 | Number of simulated $\{\theta, Y\}$ pairs; used to train $q_\phi(\theta|Y)$ |
| Number of samples | 5000 | Number of sampled $\{\theta, Y\}$ pairs; used to evaluate $q_\phi(\theta|Y)$ |

**Appendix D Inference for many observations, $Y_{Obs\_LSTM\_i}$**

A trained neural density estimator can be used to infer the parameters of an observation without the need for additional simulation runs. In this section, we extend Experiment 1 (the 'best' case) to evaluate the posterior parameter density for many synthetic observations ($Y_{Obs\_LSTM\_i}$) quickly and effectively. We use many parameter sets ($\theta_i$) of $K_s$ and $M_s$ sampled uniformly across parameter space to generate an equivalent number of synthetic observations, where *i=1, 2, ..., 441*.




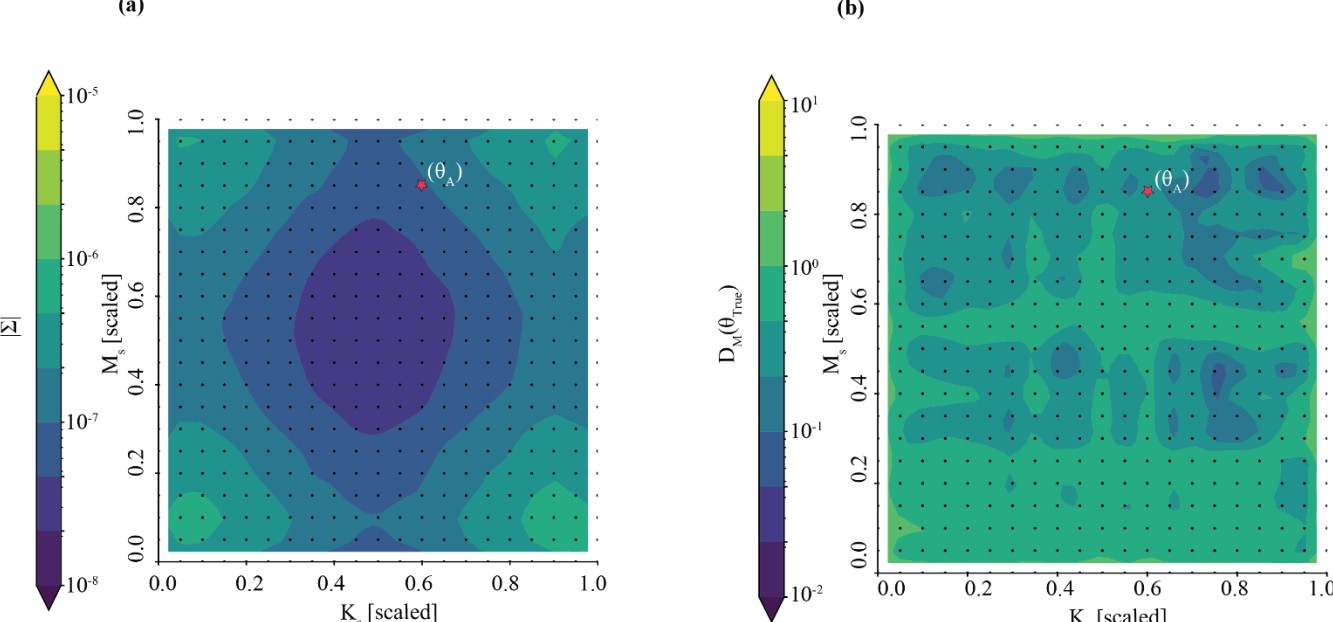

**Figure D1. Once the neural conditional density estimator is trained, it can be evaluated quickly and effectively given new data. This figure shows the performance of SBI of Mannings ($M_s$), and Hydraulic Conductivity ($K_s$) given synthetic streamflow data generated by the surrogate from across 441 locations across parameter space. Subplot (a) shows the Determinant, $|\Sigma|$ of the posterior parameter estimate, which quantifies the precision of parameter inference. Subplot (b) shows the Mahalanobis distance, $D_M(\theta_{True})$ between the inferred distribution and true parameter values, which quantifies the accuracy of inference. These values are shown across the entirety of parameter space investigated, where purple is better. The red star in subplots corresponds with benchmark location $\theta_A$ in parameter space of the analysis shown in Figure 3.**

SBI can infer the parameters from many diverse and different synthetic observations well, as shown in Figure D1. The precision of inference of the posterior parameter densities is explored in Figure D1A as a map of determinants across parameter space. Parameter inference is more precise (with a smaller determinant) in the center than at the edges of the parameter space; it is below our precision threshold of 1e-06 everywhere. Parameter inference is accurate across parameter space, as shown by the map of Mahalanobis Distance in Fig. D1B. There are some pockets of parameter space characterized by more- and less- accurate parameter inference. The structure of the Mahalanobis distances across parameter space doesn't seem to be as well-defined as that of the determinant and are likely a consequence of randomness in the initialization of the neural density estimator (confirmed by many independent trials). We note that evaluating each of the synthetic observations in Fig. D1 took only a few seconds.



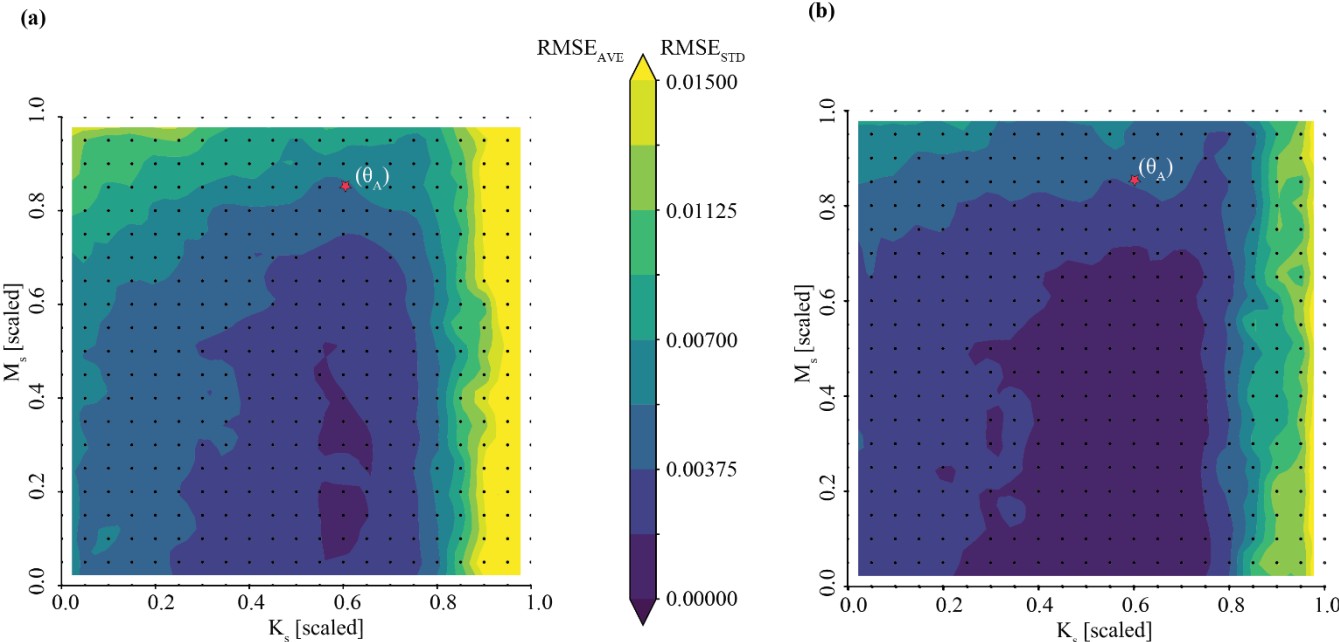

**Figure D2. Posterior predictive check for many observations: Once parameters are inferred, the posterior can be drawn (n=50) to generate probabilistic streamflow ensembles. This figure shows the performance of streamflow ensembles derived from SBI at 441 locations across parameter space. Subplot (a) shows the average of the error (RMSE$_{Ave}$) of streamflow ensembles relative to 'truth', which can be thought of as a measure of accuracy. Subplot (b) shows the standard deviation of the error (RMSE$_{std}$) of streamflow ensembles, which can be thought of as a measure of precision. Streamflow ensembles are evaluated against the 'true' synthetic streamflow time series generated by the surrogate simulator, where blue is better.**

The posterior predictive check shows that streamflow characterization is generally both precise and accurate. This required drawing a subset of parameters from *each* of the 441 posterior parameter densities represented as points in Fig. D1 and generating an ensemble of simulated streamflow time series using the surrogate simulator. The accuracy of the posterior predictions is explored in Fig. D2A as a map across parameter space. In general, the posterior predictions have an average error of less than 0.01. Accuracy is highest in the middle of the parameter space and seems to degrade towards the upper boundaries where parameters $K_s$ and $M_s$ are large. The precision of the posterior predictions is explored in Fig. D2B as a map across parameter space. In general, the posterior predictions are precise, with standard deviation of the error less than 0.01. We note that both the average and standard deviation of error increase at large parameter values, in particular large values of hydraulic conductivity. Overall, Fig. D1 and D2 show that SBI can reliably infer parameters and characterize streamflow processes for *many* streamflow observations that span the parameter space we investigated.

**Appendix E Inference on non-synthetic observations at the Taylor River**

The informal BMA methodology is suited to assessing the adequacy of model structures and configurations in the real-world case. In Figure E1, inference is conducted on the observed streamflow timeseries for water year 1995 from the Taylor River gage 09110000 (red). The figure shows the posterior predictive check with confidence intervals from standalone SBI (blue), as well as the "persistence" baseline (orange). Model configurations scoring less than persistence (defined by setting next week's predicted data equal to today's observed data) are considered not credible and assigned a weight of zero. Note that standalone SBI does not perform well relative to persistence (KGE = 0.94). The culprit is the timing of peak simulated flows, which occur on average some 44 days before the peak observation and 51 days before persistence. With no models superior to persistence, the BMA methodology returns an empty set; no model structures (LSTM surrogates) or configurations (parameter sets) yield predications that are "reasonably good". In fact, no model structures or configurations superior to persistence exist in the full space of possible combinations of M and K, as shown by the confidence intervals in grey. We emphasize to the reader that the BMA methodology results in a desirable outcome: all models identified by standalone SBI are rejected, and overconfident predictions and parameter estimates are avoided.

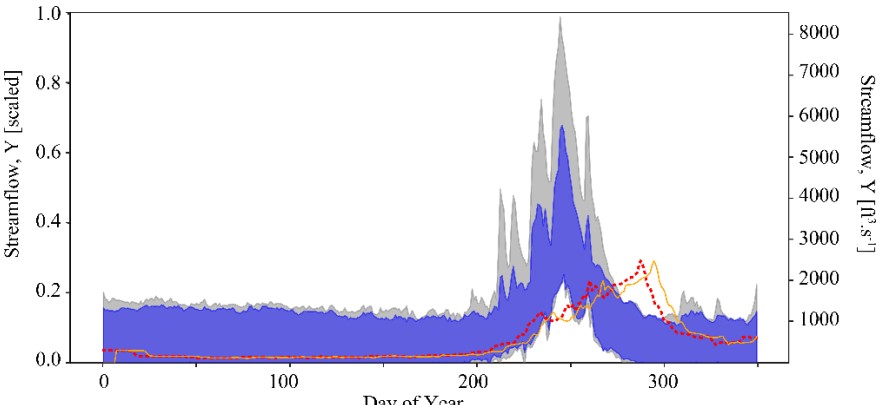

**Figure E1. Time series comparing the observed streamflow for water year 1995 (red) with the persistence baseline (orange), posterior predictive check from standalone SBI (blue), and simulations drawn from the full parameter space (gray).**

**Competing Interests**

The contact author has declared that none of the authors has any competing interests

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
