# Peer review of "Simulation-Based Inference for Parameter Estimation of Complex Watershed Simulators"

_Hydrology and Earth System Sciences, 2023_

## Referee Comment (RC2)

**Review of Manuscript**

**'Simulation-Based Inference for Parameter Estimation of Complex Watershed Simulators'**

By R. Hull et al.

Dear Editor,

I have reviewed the manuscript. My conclusions and comments are as follows:

**1. Scope**

The article is within the scope of HESS.

**2. Summary**

In their manuscript, the authors address the question of efficient parameter estimation for distributed process-based hydrological models. They suggest simulation-based inference using a surrogate model (LSTM) for the original model (Parflow) for rapid generation of parameter – simulation output data sets to support training of a second neural network to learn the joint distribution of parameters and simulation output in an nonparametric way. With their approach, they address both the intractability problem of parameter estimation (distribution cannot be properly estimated due to theoretical or computational reasons) and the epistemic uncertainty problem, here more specifically the problem of uncertainty about the correct model structure. They explore the effects of the various parts of their workflow by various virtual reality studies with different levels of simplification (experiments 1-4). They conclude that i) SBI works well if the surrogate LSTM is accurate (experiment 1), that surrogate misspecification leads to errors in parameter estimation (experiment 2), that the problem of overconfident parameter inference of experiment 2 can be partially solved by ensemble (boosted) approaches (experiment 3), and by an ensemble approach with informal weighting of the members (experiment 4).

**3. Evaluation**

This is a thoroughly conducted study on a relevant topic, reported in a complete, concise and balanced manner. In short, it was a pleasure to read. So I have only very minor specific points:

Line 63-64: I do not agree with the general claim that "DL methods are not widely used in watershed prediction due to the inadequacy of available data in representing the complex spaces of hypotheses". There are in fact many examples of DL-only or DL-conecptual hydrological modeling applications in the literature. I'd agree if the authors meant that DL methods are not widely used for distributed prediction of a large number of hard-to-observe hydrological variables. Please explain.

Line 357: For demonstration purposes, only two parameters, Manning's roughness and hydraulic conductivity are investigated. Can you say a word about how you expect the method to scale to larger number of parameters?

Line 359: 183 is not a very large ensemble. I assume this is due to the high computational effort of ParFlow? Also, can you say a word about the computational effort of the PB Model (ParFlow) vs. the NN model (LSTM)?

Line 374: "They [LSTMs] have had some use for predictions in hydrology" really is an understatement. They are in very widespread use these days. Please change.

Eq [11]: Just a comment: This could also be done by Kullback-Leibler divergence without introducing a threshold chosen by trial and error.

Eq [12]: Why is here RMSE used, instead of KGE as in Sect. 3.8?

Yours sincerely,

Uwe Ehret

---

## Author Comment (AC1)

The paper "Simulation-Based Inference for Parameter Estimation of Complex Watershed Simulators" introduces a method utilizing SBI combined with deep learning techniques to improve the calibration of process-based simulators, focusing on Manning's coefficient and hydraulic conductivity in a snowmelt-dominated catchment. It aims to address two main challenges: the computational intractability of simulating complex watershed processes for parameter estimation and the uncertainty arising from simplified model representations of these complex processes. The study performed a series of synthetic experiments to investigate the performance of the SBI approach. The study is generally well-designed and the manuscript provides a detailed explanation of the methods, experiments, and results. Overall, I have no major concerns regarding the methodology of the paper. However, I believe that some points need further clarification.

We thank the reviewer for their consideration of our work and are glad to hear that you found the experiments and manuscript to be well designed. We have provided a thorough response to all concerns below.

1) One of the basic assumptions of the presented research is the efficacy of LSTM networks as surrogates for process-based models. While the study acknowledges that LSTMs may not perfectly reproduce the behaviors, I think the LSTM model in this case study is still ideal (based on Table 3, the worst KGE can be as high as 0.77, which indicates generally good performance of the LSTM in mimicking the process-based model). This is understandable because the basin is dominated by snowmelt, which the LSTM does well due to its strong memory capacity. However, the broader applicability of the approach still needs to be discussed, especially given the known challenges LSTMs face in accurately representing hydrologic behavior in arid regions (e.g., https://doi.org/10.1029/2019WR026793).

We appreciate the comment. We did intentionally focus our work on an example where the surrogate quality is already established. Our goal in this paper is to demonstrate utility of the Simulation-Based Inference (SBI) approach and to separate this consideration from the performance of the LSTM itself (which we feel is well covered in other publications).

It is of course still true that models ought to be sufficient representations of the target system. In the situation where the surrogate does not mimic the behavior of the PB model well, it will of course result in inaccurate parameter estimates. The same goes for PB models – if they are poor predictors of observed behavior, they will perform poorly in SBI. However, we note that this problem with inadequate models is not exclusive to SBI but common to all parameter estimation methods.

A more nuanced question with respect to model adequacy is "how good is good enough?" When should – if at all – an LSTM trained on PB model representations of arid catchment conditions be used in conjunction with SBI to estimate model parameters? The manuscript details an approach to answering that question through the informal performance

weighting approach, the methodology of which is described in Sections 2.4 and 3.8, the tested with Experiment 4 as described in Section 4.4. This approach utilizes a user-defined 'limit of acceptability', which allows for exclusion of poorly performing surrogate model structures from the parameter estimation process. We are not the first to use a limit of acceptability (read any of Dr. Beven's work on GLUE), but as far as we know the first to use it in conjunction with SBI. In Experiment 4, we demonstrate that using a performance-weighted approach within the SBI framework can mitigate issues arising from mismatches between the system of interest and the surrogate model.

If a surrogate, such as one trained on conditions from an arid catchment, does not meet the defined acceptability criteria, SBI would yield no viable parameter estimates, signaling the need for reevaluation. (We explore a similar scenario in Appendix E). This outcome urges the practitioner to reconsider the assumed model structure, be it a surrogate or process-based model.

The development of robust model structures, whether they are surrogates or process-based, remains a central challenge in hydrology. The development of surrogates capable of representing spatially-distributed hydrologic systems is an active research area (see Leonarduzzi et al, 2022; and Tran et al, 2021). While this work is of course centrally important to long term model performance, we feel that topic is beyond the scope of this work. Here we seek to provide a general demonstration of the adequacy of the SBI framework. As models and surrogates improve over time the overall skill SBI will also improve. However, what we intend to show here is separate from this improvement, as we are focused on the added benefit of the SBI process itself.

We do think that this is a very important comment though and a point we want to make clearly to all readers. In response to this comment we will expand the discussion and conclusions to better highlight this distinction regarding the purpose and scope of the paper and the implications of our results as simulation platforms improve.

2) My second concern is with the terminology used throughout the description of the experimental settings, particularly the interchangeable use of the terms "observation", "simulation", and "synthetic observation". I understand that observation refers more to the simulation of Parflow, but sometimes the distinction is not so clear, forcing me to rely on context to understand their intended meanings. This ambiguity is further complicated by the term "simulator", which is used variably to refer to both "ParFlow" and the "LSTM" model (see my specific comments).

We recognize the need for clarity and consistent terminology and understand that the term 'observation' may be causing confusion as we are using it here. We thank the reviewer for the detailed comments where ambiguity in terms occurs in the manuscript and will rectify it in the next round of revisions.

3) A related point is: When I read the text discussing model misspecification, particularly lines 82-90, I initially interpreted the discussion as addressing uncertainties arising from the ParFlow model. This interpretation was influenced by the preceding context, which focuses primarily on the challenges associated with PB models. However, as I read through the manuscript, it became clear that the uncertainties referred to might actually be related to the LSTM model used in the SBI framework, since ensemble modeling is applied to LSTM models instead of ParFlow. Perhaps I misunderstood something, but it has indeed caused some confusion. If this is the case, the statement "calibrating these simulators to observed data remains a significant challenge due to several persistent issues including: …. uncertainty stemming from the choice of simplified model representations of complex natural hydrologic processes" in the Abstract would also be misleading,  as the manuscript primarily addresses uncertainties related to the LSTM surrogate simulator without adequately discussing the uncertainties inherent in the process-based (ParFlow) model itself.

We agree with the reviewer that this is a very important distinction and we see how our choice of language may have caused confusion here. Our intent was to be clear in the manuscript that the work focuses on a set of synthetic experiments meant to provide `controlled` (diagnosable) levels of uncertainty, in this case related to the relationship between a PB model and its surrogate (e.g. [23], [112-121], [299-325], [811-825]).  In the introduction and background we tried to describe the theory behind the work in a manner general enough that it applies to both the synthetic case and the real case.  In fact, our view is that the difference between model uncertainty arising from the selection of surrogate is analogous to that arising from the selection of the PB model [e.g., 839-846], as they both stem from the challenge of specifying a 'correct' model with respect to the target system. We also attempted to be clear that the task of comparing the method to 'real observations' is left for follow-up study, and describe some specific challenges related to that task (e.g. [847-865]).  That said we understand that some inconsistencies in our language (as pointed out in comment 2) may have confused some of the points we were trying to make.

We also agree with the reviewer that the abstract could be somewhat more concrete about the need for additional study of the uncertainties inherent in process-based models.

In our revision of the manuscript we will (1) correct language throughout to be sure we are explicit and consistent in all terminology relating to simulation and observations, (2) we will review the background and introduction sections to improve clarity around this comment and the first comment, (3) add a clarifying sentence to our abstract.

Specific comments are below.

Abstract: "watershed" and "catchment" are used interchangeably here and throughout the text (along with "basin"). I am not sure if the author wants to distinguish between them,

otherwise please consider a consistent word.
Will select a consistent word in the revised manuscript.

l62-64: I think the two sentences are contradictory if "watershed prediction" is identical to "streamflow prediction". Perhaps watershed prediction here refers to the prediction of hydrologic states and fluxes other than streamflow, if so, please clarify.
Will revise to `prediction of watershed variables`

l67: It would be helpful to state what the difference is between this study and Tsai's study in terms of how DL helps with parameter calibration of PB models.

We will provide a sentence or two comparison in the updated manuscript between ours and Tsai's study in terms of how DL is used.

l74: I would say that it is not always true that DL can "preserve fidelity" unless the surrogate has sufficient predictive power (see my major comment)

Will revise to describe that fidelity is not always maintained.  See also response to major comment 1.

l88 (and Sec 2.4): Which model do you mean here, the parflow, the LSTM, or the neural density estimator? This is quite confusing.

Will revise to make explicit that the statement applies to both ParFlow and the LSTM.  The statements in these sections are sufficiently general where they can apply to both ParFlow and LSTM; see response to major comment 3.

Sec 2.2: It seems that SBI itself includes deep learning (l68), but here it looks like SBI can be independent with deep learning, please clarify.

Yes this is correct, Simulation-based inference is a general class of inference that accepts/rejects simulation results based on a distance measure from observed data.  A common way of doing this in the past used Approximate Bayesian Computation (ABC), which used a frequentist approach (not involving a neural network).  More recently, a density-estimation approach that uses neural networks (deep learning) has been employed. While we get into the details of this in the body of section 2.2 we agree this might not have been clear from the start. We will revise the first section of 2.2 to clarify that there are a range of SBI approaches some of which include deep learning and some of which do not.

l279: what prior distribution is used in the study? please clarify and justify.

As described in Table C1, the prior distribution used for density estimation was uniform in Ks and Ms, which are described in Table A2 and Section 3.3. We will add a note to this part of the text pointing the reader to these later descriptions.

l299-l304: which simulator do you mean here, I assume it is the surrogate simulator (LSTM)? Because you take the result of the parflow simulator as an observation. Please clarify and unify.

Per our response to major comment 3 we agree that the distinction between simulators was not made clearly enough in the original manuscript. The bulleted list following these lines does outline clearly what is used as the simulator and what is used as observations but we agree that this point could be made more clearly in the overview. We will add a sentence here reminding the reader of what we are treating as the simulations and what we are treating as the observations for this experiment.

l316: Until I read here I am aware that you are saying the uncertainty from the surrogate simulator.

Thank you for pointing this out. We think the text we will add to the top of section 3.1 will address this concern.

l380: How was the 14 days determined?

The LSTM sequence length was determined through trial and error. As written in line 393, "We emphasize that the goal here is to produce a surrogate simulator adequate for the simulation-based inference of parameters Ks and Ms." Will specify in the updated manuscript that the sequence length was determined via trial and error.

l391: It would be nice to show the NSE or KGE value here.

Will revise to include an average KGE value for a set of samples selected uniformly across parameter space.

l391: I am curious about the applicability of the SBI approach to other basins in case the surrogate simulator does not capture the flow behavior well.

This is an interesting question. As noted in our response to major comment 1 our experiments are intentionally designed to explore the performance of SBI. Biases in simulators is a separate topic and we have tried to frame our work in a general way to

highlight the utility of SBI for many simulation platforms. However, it should of course be noted that overall skill of the method can only be as good as the underlying model that is being tuned. We tried to make this point on in our discussion section:

> "The critic might suggest that not enough was done to tailor the present analysis to real world data. We disagree on the grounds that our purpose here is to rigorously present and evaluate a method for parameter inference given well-defined constraints. The challenge of this goal is real and relevant. In fact, this work seems to show an upper bound for the performance of SBI where undiagnosed structural error exists."

And at the end of the conclusions:

> "Additional work is needed to address deeper uncertainty about the structural adequacy of the underlying physics-based model. This uncertainty often exists in watershed modeling – due to (e.g.) natural heterogeneities in the subsurface, approximations in process parameterizations, and bias in the meteorological input data – that can seldom be fully 'accounted for'. The notion of structural 'adequacy' is thus nearly always subjective (Gupta et. al, 2012). In many 'real world' applications, a calibrated estimate of the hydrologic variable (i.e., streamflow) is what watershed scientists strive for. Enhancing standalone SBI with the likelihood-weighting methodology introduced in Experiment 4 embraces this principle of subjective 'adequacy' and is broadly extendable to more complex inference problems in watershed modeling. Where no models are identified as adequate, an obvious next step is to expand the simulator to explore more and different configurations of parameters and input variables. "

As noted in our response above though we agree that this is a very important point for the understanding of our work and we will review the manuscript to make sure that this point is coming through clearly throughout the paper.

l431: Which simulator do you mean here?

The parameters in question are the physical parameters, Mannings (M) and Hydraulic Conductivity (K), from ParFlow. As noted in section 3.4, these are also inputs into the LSTM. See also footnote `1`. The statement can be applied to both the LSTM and ParFlow. We will make explicit in the text.

l471: It is not clear to me what model structure you are referring to.

The statement can be applied to both the LSTM and ParFlow. We will make explicit in the text.

l542: I find it strange to call the LSTM result a "synthetic observation".

See description of Experimental Method in Section 3.1. As noted in our response to major comment 2 we will review all of our terminology and streamline for consistency in the revised manuscript.

Figure 4: Since you have defined many "observations", please clarify which observation you mean here.

On lines l543-544 we describe this as follows: "We first infer the parameters of just one randomly selected streamflow observation, denoted with an 'A' ($Y_{Obs\_LSTM\_A}$). The set of 'benchmark' parameters ($\theta_A$) used to generate the underlying simulation are approximately 0.60 for Ks, and 0.85 for Ms. $\theta_A$ is also our benchmark in parameter space for Experiments 2."  We believe that the reader is confused by the notation `$A_{Yobs}$` used in the text, and we agree that it is incomplete and will update to `$Y_{Obs\_LSTM\_A}$`.

l559: please consider not using e-style scientific notation here (see inconsistency with l581 and l582).

Will update in the revised manuscript.

Sec 4.1.1: may remove the header if there is no "sec 4.1.2".

We agree and will remove.

l630: please justify why different settings of LSTM were used than in exp #2

Different settings were used with the intention of utilizing weaker LSTM simulators, to prevent overfitting by the neural density estimator.  We agree the text could be more clear and will remedy in the revised manuscript.

---

## Author Comment (AC2)

**Review of Manuscript**

**'Simulation-Based Inference for Parameter Estimation of Complex**

**Watershed Simulators'**

By R. Hull et al.

Dear Editor,

I have reviewed the manuscript. My conclusions and comments are as follows:

**1.  Scope**

The article is within the scope of HESS.

**2.  Summary**

In their manuscript, the authors address the question of efficient parameter estimation for distributed process-based hydrological models. They suggest simulation-based inference using a surrogate model (LSTM) for the original model (Parflow) for rapid generation of parameter – simulation output data sets to support training of a second neural network to learn the joint distribution of parameters and simulation output in an nonparametric way. With their approach,  they address both the intractability problem of parameter estimation (distribution cannot be  properly estimated due to theoretical or computational reasons) and the epistemic uncertainty problem, here more specifically the problem of uncertainty about the correct model structure. They explore the effects of the various parts of their workflow by various virtual reality studies with different levels of simplification (experiments 1-4). They conclude that i) SBI works well if the surrogate LSTM is accurate (experiment 1), that surrogate misspecification leads to errors in parameter estimation (experiment 2), that the problem of overconfident parameter inference of experiment 2 can be partially solved by ensemble (boosted) approaches (experiment 3), and by an ensemble approach with informal weighting of the members (experiment 4).

**3.  Evaluation**

This is a thoroughly conducted study on a relevant topic, reported in a complete, concise and balanced manner. In short, it was a pleasure to read. So I have only very minor specific points:

We thank the reviewer for their thorough review of our work and for their positive evaluation. We have address all of the specific comments below.

Line 63-64: I do not agree with the general claim that "DL methods are not widely used in watershed prediction due to the inadequacy of available data in representing the complex spaces of hypotheses". There are in fact many examples of DL-only or DL-conecptual hydrological modeling applications in the literature. I'd agree if the authors meant that DL methods are not widely used for distributed prediction of a large number of hard-to-observe hydrological variables. Please explain.

We agree with the reviewer's comment that DL applications have become more common in recent years.  Our intent was to emphasize they are not yet widely used to predict distributed variables in hydrology.  We will revise the paragraph to emphasize that the novelty is in distributed prediction, in this case using an emulator of ParFlow, a distributed physically-based model.

Line 357: For demonstration purposes, only two parameters, Manning's roughness and hydraulic conductivity are investigated. Can you say a word about how you expect the method to scale to larger number of parameters?

We briefly address this question in discussion [lines 847-865].  We will lengthen this discussion in the revision to better cover the following points:

Expanding the model, both in terms of the number and distribution of parameters, is essential to finding adequate representations of real hydrological processes.  SBI is well-suited for inference in high-dimensional space relative to some approaches, and has had many adaptations (Cranmer, 2021).  As with any approach to inference, scaling to a greater number of parameters will bump into computational constraints.  Those constraints come from the cost of simulation (i.e. ParFlow), and the cost of inference (i.e. neural density estimation).  In our study, the cost of simulation from ParFlow is high, and this has a compounding effect on the cost of inference.  Utilizing a surrogate can in some ways reduce the cost of inference, by reducing the need to resort to ParFlow, but as we show this comes at a tradeoff of accurate parameter estimates if the surrogate is not adequate.  Focusing inference on the most informative parts of higher-dimensional parameter space is important if SBI is conducted directly with a costly simulator.  Papamarkarios' early work with SBI developed sequential neural sampling techniques, which might be less wasteful than other approaches to sampling parameter space (i.e. Papamakarios et al., 2018; Lueckmann et al., 2017; Greenberg et al., 2019).  Lastly is the option of compressing or reducing the dimensionality, which could be important for the case of estimating distributed parameters.  The topic of compression and SBI is explored by Asling, 2019.

Line 359: 183 is not a very large ensemble. I assume this is due to the high computational effort of ParFlow? Also, can you say a word about the computational effort of the PB Model (ParFlow) vs. the NN model (LSTM)?

The reviewer is correct that this is primarily due to the cost of the process based model.  As noted in the previous response, this is the largest part of our computational demand.  The computational cost of training of the LSTM is many orders of magnitude smaller compared to the cost of ParFlow simulations.  Once trained, simulation from the LSTM is essentially free.  In the revised manuscript we will include specific details related to the cost of ParFlow vs. LSTM.

Line 374: "They [LSTMs] have had some use for predictions in hydrology" really is an understatement. They are in very widespread use these days. Please change.

At the time of drafting (2021), they were still somewhat nascent.  We will update in the revised manuscript.

Eq [11]: Just a comment: This could also be done by Kullback-Leibler divergence without introducing a threshold chosen by trial and error.

Thanks for the comment.  We could see how this approach will be advantageous in future evaluations of SBI.

Eq [12]: Why is here RMSE used, instead of KGE as in Sect. 3.8?

As referenced in the text from [490], we choose to use the Kling Gupta Efficiency (KGE; Gupta et al., 2009) as the likelihood metric for its utility and history rainfall-runoff model assessment. We appreciate the reviewer for noting alternatively that the RMSE was selected to evaluate the posterior prediction. However, using KGE instead for the posterior predictive check would not change the results and conclusions in the manuscript.

Yours sincerely,

Uwe Ehret